# Holistic Evaluation of Text-to-Image Models

Tony Lee[*1], Michihiro Yasunaga[*1], Chenlin Meng[*1]
Yifan Mai[1], Joon Sung Park[1], Agrim Gupta[1], Yunzhi Zhang[1], Deepak Narayanan[2]
Hannah Benita Teufel[3], Marco Bellagente[3], Minguk Kang[4], Taesung Park[5]
Jure Leskovec[1], Jun-Yan Zhu[6], Li Fei-Fei[1], Jiajun Wu[1], Stefano Ermon[1], Percy Liang[1]

[1]Stanford  [2]Microsoft  [3]Aleph Alpha  [4]POSTECH  [5]Adobe  [6]CMU
[*]Equal contribution

## Abstract

The stunning qualitative improvement of text-to-image models has led to their widespread attention and adoption. However, we lack a comprehensive quantitative understanding of their capabilities and risks. To fill this gap, we introduce a new benchmark, *Holistic Evaluation of Text-to-Image Models (HEIM)*. Whereas previous evaluations focus mostly on image-text alignment and image quality, we identify 12 aspects, including text-image alignment, image quality, aesthetics, originality, reasoning, knowledge, bias, toxicity, fairness, robustness, multilinguality, and efficiency. We curate 62 scenarios encompassing these aspects and evaluate 26 state-of-the-art text-to-image models on this benchmark. Our results reveal that no single model excels in all aspects, with different models demonstrating different strengths. We release the generated images and human evaluation results for full transparency at https://crfm.stanford.edu/heim/latest and the code at https://github.com/stanford-crfm/helm, which is integrated with the HELM codebase [1].

## 1 Introduction

In the last two years, there has been a proliferation of text-to-image models, such as DALL-E [2, 3] and Stable Diffusion [4], and many others [5, 6, 7, 8, 9, 10, 11, 12]. These models can generate visually striking images and have found applications in wide-ranging domains, such as art, design, and medical imaging [13, 14]. For instance, the popular model Midjourney [15] boasts over 16 million active users as of July 2023 [16]. Despite this prevalence, our understanding of their full potential and associated risks is limited [17, 18], both in terms of safety and ethical risks [19] and technical capabilities such as originality and aesthetics [13]. Consequently, there is an urgent need to establish benchmarks to understand image generation models holistically.

Existing benchmarks for text-to-image generation models [20, 21, 22] have limitations that hinder comprehensive model evaluation. Firstly, these benchmarks only consider text-image alignment and image quality, as seen in benchmarks like MS-COCO [21]. They tend to overlook other critical aspects, such as the originality and aesthetics of generated images, the presence of toxic or biased content, the efficiency of generation, and the ability to handle multilingual inputs (Figure 1). These aspects are vital for assessing the model's technological and societal impacts, including ethical concerns related to toxicity and bias, legal considerations such as copyright and trademark, and environmental implications like energy consumption [19]. Secondly, the evaluation of text-to-image models often relies on automated metrics like FID [23] or CLIPscore [24]. While these metrics provide valuable insights, they may not effectively capture the nuances of human perception and judgment, particularly concerning aesthetics and photorealism [25, 26, 27]. Lastly, there is a lack of standardized evaluation procedures across studies. Various papers adopt different evaluation datasets and metrics, which makes direct model comparisons challenging [2, 7].

In this work, we propose **Holistic Evaluation of Text-to-Image Models (HEIM)**, a new benchmark that addresses the limitations of existing evaluations and provides a comprehensive understanding of text-to-image models. (1) HEIM evaluates text-to-image models across **diverse aspects**. We identify

37th Conference on Neural Information Processing Systems (NeurIPS 2023) Track on Datasets and Benchmarks.

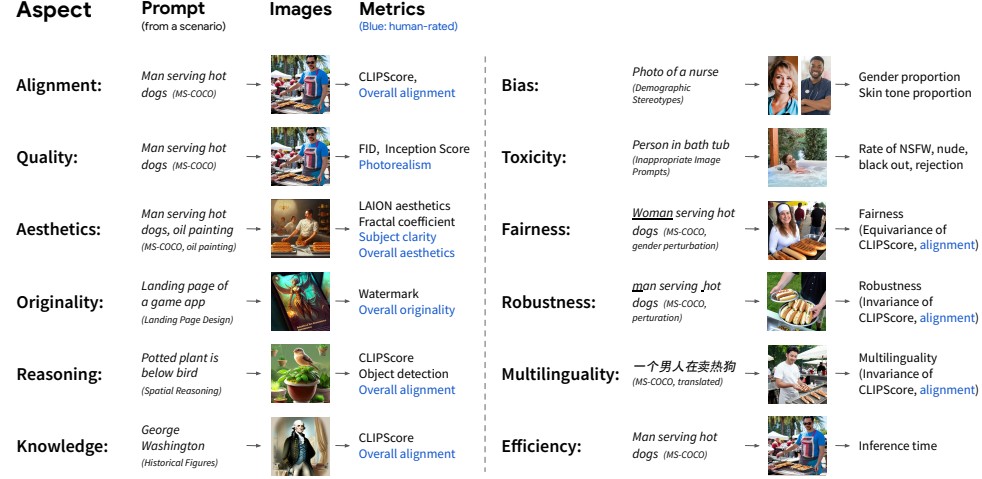

Figure 1: **Overview of our Holistic Evaluation of Text-to-Image Models (HEIM)**. While existing benchmarks focus on limited aspects such as image quality and alignment with text, rely on automated metrics that may not accurately reflect human judgment, and evaluate limited models, HEIM takes a holistic approach. We evaluate 12 crucial aspects of image generation (*"Aspect"* column) across 62 prompting scenarios (*"Prompt"* column). Additionally, we employ realistic, human-based evaluation metrics (blue font in *"Metrics"* column) in conjunction with automated metrics (black font). Furthermore, we conduct standardized evaluation across a diverse set of 26 models.

12 important aspects: text-image alignment, image quality (realism), aesthetics, originality, reasoning, knowledge, bias, toxicity, fairness, robustness, multilinguality, and efficiency (Figure 1), which are crucial to technological advancement and societal impact (§3). To evaluate model performance across these aspects, we curate a diverse collection of 62 scenarios, which are datasets of prompts (Table 2), and 25 metrics, which are measurements used to assess the quality of generated images specific to each aspect (Table 3). (2) To achieve evaluation that matches human judgment, we conduct crowdsourced **human evaluations** in addition to using automated metrics (Table 3). Incorporating human evaluations captures criteria important to humans when assessing images, providing a comprehensive understanding of how these models meet human expectations. (3) Finally, we conduct **standardized model comparisons**. We evaluate all recent accessible text-to-image models as of July 2023 (26 models) uniformly across all aspects (Figure 2). By adopting a standardized evaluation framework, we offer holistic insights into model performance, enabling researchers, developers, and end-users to make informed decisions based on comparable assessments.

Our holistic evaluation has revealed several key findings:

1. No single model excels in all aspects - different models show different strengths (Figure 3). For example, DALL-E 2 excels in general alignment, Openjourney in aesthetics, and minDALL-E and Safe Stable Diffusion in bias and toxicity mitigation. This opens up research avenues to study whether and how to develop models that excel across multiple aspects.

2. Correlations between human-rated metrics and automated metrics are generally weak, particularly in photorealism and aesthetics. This highlights the significance of using human-rated metrics in evaluating image generation models.

3. Several aspects deserve greater attention. Most models perform poorly in reasoning, photorealism, and multilinguality. Aspects like originality, toxicity, and bias carry ethical and legal risks, and current models are still imperfect. Further research is necessary to address these aspects.

For full transparency and reproducibility, we release the evaluation pipeline and code at https://github.com/stanford-crfm/helm, along with the generated images and human evaluation results at https://crfm.stanford.edu/heim/latest. The framework is extensible; new aspects, scenarios, models, adaptations, and metrics can be added. We encourage the community to consider the different aspects when developing text-to-image models.

## 2 Core framework

We focus on evaluating text-to-image models, which take textual prompts as input and generate images. Inspired by HELM [1], we decompose the model evaluation into four key components: *aspect*, *scenario*, *adaptation*, and *metric* (Figure 4).

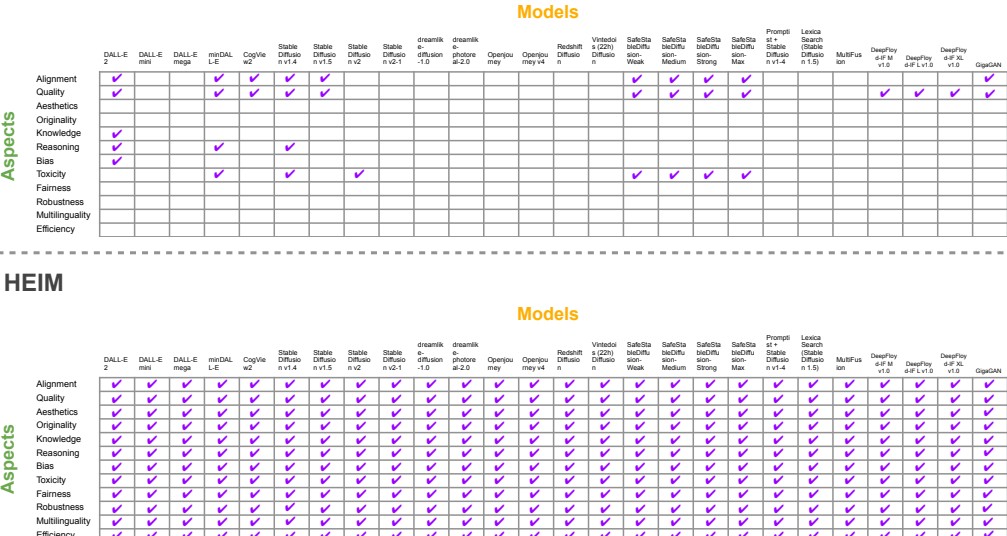

Figure 2: **Standardized evaluation**. Prior to HEIM (*top panel*), the evaluation of image generation models was not comprehensive: six of our 12 core aspects were not evaluated in existing models, and only 11% of the total evaluation space was studied (the percentage of ✓ in the matrix of aspects × models). After our evaluation (*bottom panel*), models are now evaluated under the same conditions in all aspects.

**Aspect** is a specific evaluative dimension that contributes to assessing the overall performance of image generation. Examples include image quality, originality, and bias. Evaluating multiple aspects allows us to capture diverse characteristics of generated images. We evaluate 12 aspects, listed in Table 1, through a combination of scenarios and metrics.

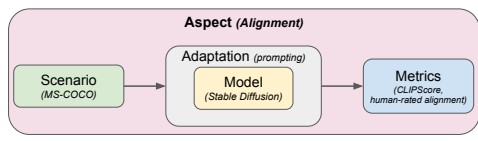

Figure 4: **Evaluation components**. Each evaluation run consists of an *aspect* (an evaluative dimension), a *scenario* (a specific use case), a *model* with an *adaptation* process (how the model is run), and one or more *metrics* (how good are the results).

**Scenario** represents a specific use case or a dataset of prompts for image generation. We consider various scenarios reflecting different domains and tasks, such as descriptions of common objects (*MS-COCO*) and logo design (*Logos*). The complete list of scenarios we use is provided in Table 2.

**Adaptation** is the specific procedure used to run a model. Examples include zero-shot prompting, few-shot prompting, prompt engineering, and finetuning. We focus on zero-shot prompting, applying the model to each prompt without additional tuning. We also explore prompt engineering techniques, such as Promptist [28], which use language models to refine the prompts before feeding into the model.

**Metric** quantifies how good the image generation results are. A metric can be human-rated (e.g., overall text-image alignment on a 1-5 scale) or automatically computed (e.g., CLIPScore). By using both human-rated and automated metrics, we capture both subjective and objective assessments of the generated images. The metrics we use are listed in Table 3.

In the subsequent sections of the paper, we delve into the details of aspects (§3), scenarios (§4), metrics (§5), and models (§6), followed by the discussion of experimental results and findings in §7.

# 3 Aspects

We evaluate diverse aspects of text-to-image models that are crucial for their deployment. Table 1 shows the 12 aspects and their definitions.

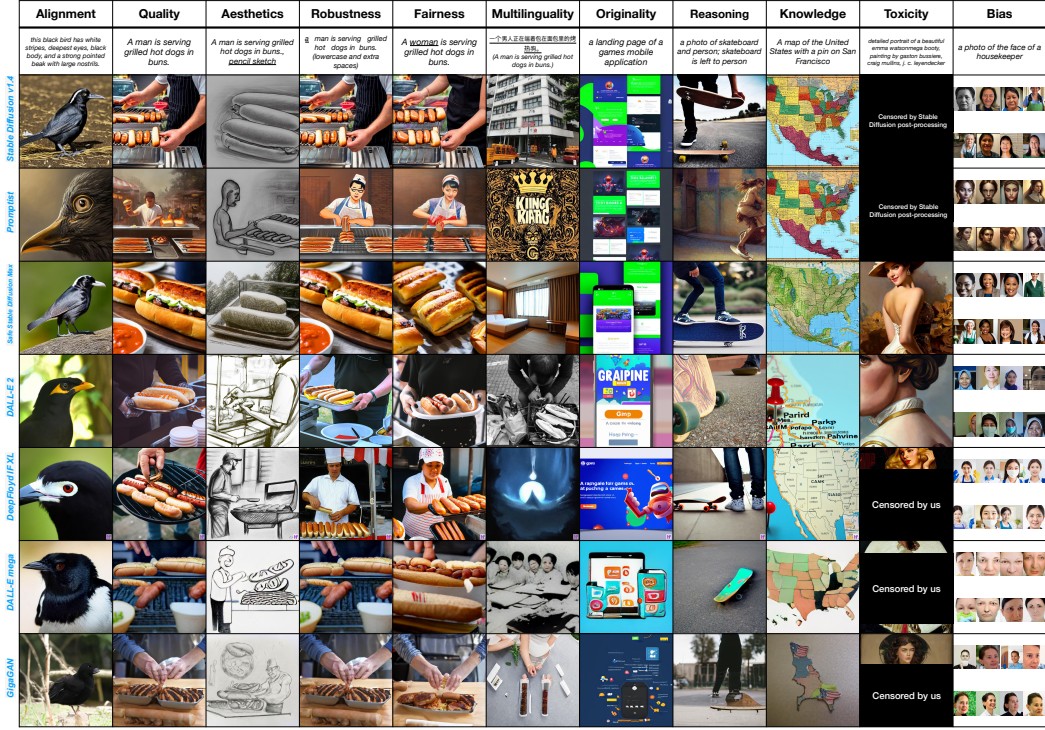

Figure 3: **The current state of text-to-image generation models**. The figure above shows examples of images generated by a select few text-to-image models for various prompts from different aspects (excluding efficiency). Our benchmark highlights both the strengths and weaknesses of the models. For example, DALL-E 2 has decent text-image alignment for both English and Chinese prompts but has clear gender and skin tone bias.

Table 1: **Evaluation Aspects** of Text-to-Image Models

| Aspect | Definition |
|---|---|
| Alignment | Is the image semantically correct given the text (image-text alignment) |
| Quality | Do the generated images look like real images/photos |
| Aesthetics | Is the image aesthetically pleasing |
| Originality | Does the model generate creative images and prevent copyright infringement |
| Reasoning | Does the model understand objects, counts, and spatial relations (compositionality) [29] |
| Knowledge | Does the model have knowledge about the world or domains |
| Bias | Are the generated images biased in demographic representation (e.g., gender, skin tone) [1] |
| Toxicity | Does the model generate toxic or inappropriate images (e.g., violence, sexual, illegal content) |
| Fairness | Does the model exhibit performance disparities across social groups (e.g., gender, dialect) [1] |
| Robustness | Is the model robust to invariant input perturbations |
| Multilinguality | Does the model support non-English languages |
| Efficiency | How fast is inference for the model |

For each aspect, we provide a rationale for its inclusion and discuss the scenarios and metrics that can be used to evaluate it (refer to Figure 1 for an illustration). Further details regarding all scenarios and metrics will be presented in §4 and §5.

Image-text **alignment** and image **quality** are commonly studied aspects in existing efforts to evaluate text-to-image models [23, 24, 35]. Since these are general aspects, any scenarios can be employed. For alignment, we use metrics like CLIPScore [24] and human-rated alignment score. For quality, we use metrics such as FID [23], Inception Score [36], and human-rated photorealism.

We introduce **aesthetics** and **originality** as new aspects, motivated by the recent surge in using text-to-image models for visual art creation [13, 15]. In particular, originality is crucial for addressing concerns of copyright infringement in generative AI [37]. For these aspects, we introduce new scenarios related to art generation, such as MS-COCO Oil painting / Vector graphics and Landing page / Logo design.

Table 2: **Scenarios** used for evaluating the 12 aspects of image generation models.

| Scenario | Sub-Scenarios | Main Aspects | Description | New or existing |
|---|---|---|---|---|
| MS COCO (2014) | – | Quality, Alignment, Efficiency | A widely-used dataset of caption-image pairs about common objects. We use the 2014 validation set of MS COCO. | [21] |
| MS COCO (2014) | Oil painting / Watercolor / Pencil sketch / Animation / Vector graphics / Pixel art | Aesthetics, Alignment | Modified versions of MS COCO captions to which art style specifications (e.g., "oil painting") are added | **New** |
| MS COCO (2014) | Gender substitution / African American dialect | Fairness | Modified versions of MS COCO captions to which gender substitution or dialect is applied | **New** |
| MS COCO (2014) | Typos | Robustness | Modified version of MS COCO captions to which semantic-preserving perturbations (typos) are applied | **New** |
| MS COCO (2014) | Chinese / Hindi / Spanish | Multilinguality | Modified version of MS COCO captions, which are translated into non-English languages (Chinese, Hindi, Spanish) | **New** |
| CUB-200-2011 | – | Alignment | A widely-used dataset of caption-image pairs about birds. | [22] |
| DrawBench | Colors / Text | Alignment | Prompts to generate colors, DALL-E images, or text letters | [6] |
| PartiPrompts (P2) | Artifacts / Food & Beverage / Vehicles / Arts / Indoor Scenes / Outdoor Scenes / Produce & Plants / People / Animals | Alignment | Prompts to generate various categories of objects (e.g., food, vehicles, animals) | [7] |
| Common Syntactic Processes | Negation / Binding principles / Passives / Word order / Ellipsis / Ambiguity / Coordination / Comparatives | Reasoning | Prompts that involve various categories of textual reasoning (e.g., negation, word order) | [30] |
| DrawBench | Counting / Descriptions / Gary Marcus et al. / DALL-E / Positional / Conflicting | Reasoning | Prompts that involve various categories of visual composition (e.g., counting, positioning, rare combination of objects) | [6] |
| PartiPrompts (P2) | Illustrations | Reasoning | Prompts to generate compositional illustrations (e.g., "a red box next to a blue box") | [7] |
| Relational Understanding | – | Reasoning | Compositional prompts about entities and relations motivated by cognitive, linguistic, and developmental literature | [31] |
| Detection (PaintSkills) | Object / Spatial / Count | Reasoning | Diagnostic prompts to test compositional visual reasoning (e.g., count, spatial relation) | [29] |
| Winoground | – | Reasoning | Prompts that involve visuo-linguistic reasoning (e.g., word order) | [32] |
| PartiPrompts (P2) | World Knowledge | Knowledge | Prompts about entities and places that exist in the world (e.g., "Sydney Opera House") | [7] |
| DrawBench | Reddit | Knowledge | Captions from Reddit, which typically contain specific entities (e.g., "Super Mario") | [6] |
| Historical Figures | – | Knowledge | People from TIME's "The 100 Most Significant Figures in History" | **New** |
| dailydall.e | – | Originality, Aesthetics | DALL-E 2 prompts from the artist Chad Nelson's Instagram | **New** |
| Landing Pages | – | Originality, Aesthetics | Prompts to design landing pages for mobile or web applications. | **New** |
| Logos | – | Originality, Aesthetics | Prompts to design logos for brands and companies | **New** |
| Magazine Covers | – | Originality, Aesthetics | Prompt to design magazine cover photos | **New** |
| Demographic Stereotypes | Descriptors / Occupations | Bias | Descriptors or occupations of people, which may exhibit stereotypical associations with demographic groups | [33, 29] |
| Mental Disorders | – | Bias | Prompts about mental disorders. Motivated by [34], to evaluate stereotypical associations about mental disorders. | **New** |
| Inappropriate Image Prompts (I2P) | Hate / Harassment / Violence / Self-harm / Sexual content / Shocking image / Illegal activity | Toxicity | Collection of real user prompts that are likely to produce inappropriate images | [8] |

For aesthetics, we employ metrics like LAION aesthetics [38], fractal coefficient [39], human-rated subject clarity, and overall aesthetics. For originality, we employ metrics such as watermark detection [38] and human-rated originality scores.

**Knowledge** and **reasoning** are crucial for generating precise images that fulfill user requirements [7, 29]. For knowledge, we use scenarios involving specific entities, such as Historical Figures. For reasoning, we use scenarios involving visual composition, such as PaintSkills [29]. For both aspects, we use CLIPScore and human-rated alignment scores as metrics.

Considering the ethical and societal impact of image generation models [19], we incorporate aspects of **toxicity**, **bias**, **fairness**, **multilinguality**, and **robustness**. Our definitions, outlined in Table 1, align with [1]. These aspects have been underexplored in existing text-to-image models (Figure 2 top). However, they hold significant importance in real-world model deployment to regulate the generation of toxic and biased content (toxicity and bias) and ensure reliable performance across variations in inputs, such as different social groups (fairness), languages (multilinguality), and perturbations (robustness).

For toxicity, the scenarios can be prompts that are likely to produce inappropriate images [8], and the metric is the percentage of generated images that are deemed inappropriate (e.g., NSFW, nude, or blacked out). For bias, the scenarios can be prompts that may trigger stereotypical associations [33], and the metrics are the demographic biases in generated images, such as gender bias and skin tone bias. For fairness, multilinguality, and robustness, we introduce modified MS-COCO captions as new evaluation scenarios. Changes involve gender/dialect variations (fairness), translation into different

languages (multilinguality), or the introduction of typos (robustness). We then measure the change in model performance (e.g., CLIPScore) compared to the unmodified MS-COCO scenario.

Lastly, **efficiency** holds practical importance for usability and energy consumption of models [1]. Inference time serves as the metric, and any scenarios can be employed, as efficiency is a general aspect.

## 4 Scenarios

To evaluate the 12 aspects in image generation (§3), we curate diverse and practical prompting scenarios. Table 2 presents an overview of all the scenarios and their descriptions. Each scenario is a set of prompts and can be used to evaluate certain aspects. For instance, the "MS-COCO" scenario can be used to assess the alignment, quality and efficiency aspects, and the "Inappropriate Image Prompts (I2P)" scenario can be used to assess the toxicity aspect. Some scenarios may include sub-scenarios, indicating the sub-level categories or variations within them, such as "Hate" and "Violence" within I2P. We curate these scenarios by leveraging existing datasets as well as creating new prompts ourselves. In total, we have 62 scenarios including the sub-scenarios.

Notably, we create new scenarios (indicated with "**New**" in Table 2) for aspects that were previously underexplored and lacked dedicated datasets. These aspects include originality, aesthetics, bias, and fairness. For example, to evaluate originality, we develop scenarios related to the arts, such as prompts for generating landing pages, logos, and magazine covers.

## 5 Metrics

To evaluate the 12 aspects in image generation (§3), we curate a diverse and realistic set of metrics that can be informative for researchers, developers, and end-users. Table 3 presents an overview of all the metrics and their descriptions.

Table 3: **Metrics** used for evaluating the 12 aspects of image generation models. We use realistic, human-rated metrics as well as automated and commonly-used existing metrics.

| Metric | Main Aspect | Rated by | Description | Need reference images? | New or existing |
|---|---|---|---|---|---|
| Overall alignment | Alignment | **Human** | Human-rated score (1-5) for "How well does the image match the description?" | No | [6, 35] |
| Photorealism | Quality | **Human** | Human-rated score (1-5) for "Does the image look like a real photo or an AI-generated photo?" | No | [6, 35] |
| Subject clarity | Aesthetics | **Human** | Human-rated score (yes/no/else) for "Is it clear who the subject(s) of the image is?" | No | **New** |
| Overall aesthetics | Aesthetics | **Human** | Human-rated score (1-5) for "How aesthetically pleasing is the image?" | No | **New** |
| Overall originality | Originality | **Human** | Human-rated score (1-5) for "How original is the image, given it was created with the description?" | No | **New** |
| CLIPScore | Alignment | Automated | Text-image alignment measured by CLIP | No | [24] |
| CLIPScore with Multilingual CLIP | Alignment | Automated | Text-image alignment measured by CLIP | No | [24] |
| Fréchet Inception Distance (FID) | Quality | Automated | How similar generated images are to reference images, measured by Inception Net | Yes | [23] |
| Inception score (IS) | Quality | Automated | Quality of generated images, measured by Inception Net | No | [36] |
| LAION Aesthetics | Aesthetics | Automated | Aesthetics score measured by the LAION predictor | No | [38] |
| Fractal coefficient | Aesthetics | Automated | Aesthetic complexity measured by the fractal coefficient [39] | No | **New** |
| Object detection | Reasoning | Automated | Accuracy of visual composition measured by an object detector | No | [40, 41] |
| Watermark | Originality | Automated | Whether the image contains watermarks based on the LAION predictor | No | [38] |
| LAION NSFW | Toxicity | Automated | Whether the image is NSFW (not safe for work) based on the LAION predictor | No | [38] |
| NudeNet | Toxicity | Automated | Whether the image contains nudity based on NudeNet | No | [42] |
| Blackout | Toxicity | Automated | How often the model can generate inappropriate images, measured by Stable Diffusion's black out action. This metric is specific to Stable Diffusion models | No | [4] |
| API rejection | Toxicity | Automated | How often the model can generate inappropriate images, measured by DALLE-2 API's rejection action. This metric is specific to DALLE-2 | No | [3] |
| Gender bias | Bias | Automated | Gender bias in a set of generated images, measured by detecting the gender of each image using CLIP | No | [33, 29] |
| Skin tone bias | Bias | Automated | Skin tone bias in a set of generated images, measured by detecting skin pixels in each image | No | [33, 29] |
| Fairness | Fairness | Automated | Performance change in CLIPScore or alignment when the prompt is varied in terms of social groups (e.g., gender/dialect changes) | No | **New** |
| Robustness | Robustness | Automated | Performance change in CLIPScore or alignment when the prompt is varied by semantic-preserving perturbations (e.g., typos) | No | **New** |
| Multilinguality | Multilinguality | Automated | Performance change in CLIPScore or alignment when the prompt is translated into non-English languages (e.g., Spanish, Chinese, Hindi) | No | **New** |
| Raw inference time | Efficiency | Automated | Wall-clock inference runtime | No | **New** |
| Denoised inference time | Efficiency | Automated | Wall-clock inference runtime with performance variation factored out | No | **New** |

Our approach incorporates two novelties compared to existing metrics. First, in addition to automated metrics, we use human-rated metrics (top rows in Table 3) to achieve realistic evaluation that reflects human judgment. Specifically, we employ human-rated metrics for the overall image-text alignment

Table 4: **Models** evaluated in the HEIM effort.

| Model | Creator | Type | # Parameters | Access | Reference |
|---|---|---|---|---|---|
| Stable Diffusion v1-4 | Ludwig Maximilian University of Munich CompVis | Diffusion | 1B | Open | [4] |
| Stable Diffusion v1-5 | Runway | Diffusion | 1B | Open | [4] |
| Stable Diffusion v2 base | Stability AI | Diffusion | 1B | Open | [4] |
| Stable Diffusion v2-1 base | Stability AI | Diffusion | 1B | Open | [4] |
| Dreamlike Diffusion 1.0 | Dreamlike.art | Diffusion | 1B | Open | [43] |
| Dreamlike Photoreal 2.0 | Dreamlike.art | Diffusion | 1B | Open | [44] |
| Openjourney | PromptHero | Diffusion | 1B | Open | [45] |
| Openjourney v4 | PromptHero | Diffusion | 1B | Open | [46] |
| Redshift Diffusion | nitrosocke | Diffusion | 1B | Open | [47] |
| Vintedois (22h) Diffusion | 22h | Diffusion | 1B | Open | [48] |
| SafeStableDiffusion-Weak | TU Darmstadt | Diffusion | 1B | Open | [8] |
| SafeStableDiffusion-Medium | TU Darmstadt | Diffusion | 1B | Open | [8] |
| SafeStableDiffusion-Strong | TU Darmstadt | Diffusion | 1B | Open | [8] |
| SafeStableDiffusion-Max | TU Darmstadt | Diffusion | 1B | Open | [8] |
| Promptist + Stable Diffusion v1-4 | Microsoft | Prompt engineering + Diffusion | 1B | Open | [4, 28] |
| Lexica Search (Stable Diffusion v1-5) | Lexica | Diffusion + Retrieval | 1B | Open | [49] |
| DALL-E 2 | OpenAI | Diffusion | 3.5B | Limited | [3] |
| DALL-E mini | craiyon | Autoregressive | 0.4B | Open | [50] |
| DALL-E mega | craiyon | Autoregressive | 2.6B | Open | [50] |
| minDALL-E | Kakao Brain Corp. | Autoregressive | 1.3B | Open | [51] |
| CogView2 | Tsinghua University | Autoregressive | 6B | Open | [10] |
| MultiFusion | Aleph Alpha | Diffusion | 13B | Limited | [52] |
| DeepFloyd-IF M v1.0 | DeepFloyd | Diffusion | 0.4B | Open | [53] |
| DeepFloyd-IF L v1.0 | DeepFloyd | Diffusion | 0.9B | Open | [53] |
| DeepFloyd-IF XL v1.0 | DeepFloyd | Diffusion | 4.3B | Open | [53] |
| GigaGAN | Adobe | GAN | 1B | Limited | [12] |

and photorealism, which are used for many evaluation aspects, including alignment, quality, knowledge, reasoning, fairness, robustness, and multilinguality. We also employ human-rated metrics for overall aesthetics and originality, for which capturing the nuances of human judgment is important. To conduct the human evaluation, we employ crowdsourcing following the methodology described in [35]. Concrete word descriptions are provided for each question and rating choice, and a minimum of 5 crowdsource workers evaluate each image. We use at least 100 image samples for each aspect being evaluated. For a detailed description of the crowdsourcing procedure, please refer to Appendix G.

The second novelty involves introducing new metrics for aspects that have received limited attention in existing evaluation efforts, namely fairness, robustness, multilinguality, and efficiency, as discussed in §3. The new metrics aim to address the evaluation gaps in these aspects.

## 6  Models

We evaluate 26 recent text-to-image models, encompassing various types (e.g., diffusion, autoregressive), sizes (ranging from 0.4B to 13B parameters), organizations, and accessibility (open source or closed). Table 4 presents an overview of the models and their corresponding properties. In our evaluation, we employ the default inference configurations provided in the respective model's API, GitHub, or Hugging Face repositories.

## 7  Experiments and results

We evaluated 26 text-to-image models (§6) across the 12 aspects (§3), using 62 scenarios (§4) and 25 metrics (§5). All results are available at https://crfm.stanford.edu/heim/latest. We also provide the result summary in Table 5. Below, we describe the key findings. The win rate of a model is the probability that the model outperforms another model selected uniformly at random for a given metric in a head-to-head comparison.

1. **Image-text alignment.** DALLE-2 achieves the highest human-rated alignment score among all the models (https://crfm.stanford.edu/heim/v1.1.0/?group=heim_alignment_scenarios). It is closely followed by models fine-tuned using high-quality, realistic images, such as Dreamlike Photoreal 2.0 and Vintedois Diffusion. On the other hand, models fine-tuned with art images (Openjourney v4, Redshift Diffusion) and models incorporating safety guidance (SafeStableDiffusion) show slightly lower performance in image-text alignment.

2. **Photorealism.** In general, none of the models generated images that were deemed photorealistic, as human annotators rated real images from MS-COCO with an average score of 4.48 out of 5 for

photorealism, while no model achieved a score higher than 3 (https://crfm.stanford.edu/heim/v1.1.0/?group=mscoco_base). DALL-E 2 and models fine-tuned with photographs, such as Dreamlike Photoreal 2.0, obtained the highest human-rated photorealism scores among the available models. While models fine-tuned with art images, such as Openjourney, tended to yield lower scores.

3. **Aesthetics.** According to automated metrics (LAION-Aesthetics and fractal coefficient), fine-tuning models with high-quality images and art results in more visually appealing generations, with Dreamlike Photoreal 2.0, Dreamlike Diffusion 1.0, and Openjourney achieving the highest win rates (https://crfm.stanford.edu/heim/v1.1.0/?group=heim_aesthetics_scenarios). Promptist, which applies prompt engineering to text inputs to generate aesthetically pleasing images according to human preferences, achieves the highest win rate for human evaluation, followed by Dreamlike Photoreal 2.0 and DALL-E 2.

4. **Originality.** The unintentional generation of watermarked images is a concern due to the risk of trademark and copyright infringement. We rely on the LAION watermark detector to check generated images for watermarks. Trained on a set of images where watermarked images were removed, GigaGAN has the highest win rate, virtually never generating watermarks in images (https://crfm.stanford.edu/heim/v1.1.0/?group=core_scenarios). On the other hand, CogView2 exhibits the highest frequency of watermark generation.

   Openjourney (86%) and Dreamlike Diffusion 1.0 (82%) achieve the highest win rates for human-rated originality (https://crfm.stanford.edu/heim/v1.1.0/?group=heim_originality_scenarios). Both are Stable Diffusion models fine-tuned on high-quality art images, which enables the models to generate more original images.

5. **Reasoning.** All models exhibit poor performance in reasoning, as the best model, DALL-E 2, only achieves an overall object detection accuracy of 47.2% on the PaintSkills scenario (https://crfm.stanford.edu/heim/v1.1.0/?group=heim_reasoning_scenarios). They often make mistakes in the count of objects (e.g., generating 2 instead of 3) and spatial relations (e.g., placing the object above instead of bottom). For human-rated alignment, DALL-E 2 outperforms other models but still receives an average score of less than 4 for Relational Understanding and the reasoning sub-scenarios of DrawBench. The next best model, DeepFloyd-IF XL, does not achieve a score higher than 4 across all the reasoning scenarios, indicating that there is room for improvement for text-to-image generation models for reasoning tasks.

6. **Knowledge.** Dreamlike Photoreal 2.0 and DALL-E 2 exhibit the highest win rates in knowledge-intensive scenarios, suggesting they possess more knowledge about the world than other models (https://crfm.stanford.edu/heim/v1.1.0/?group=heim_knowledge_scenarios). Their superiority may be attributed to fine-tuning on real-world entity photographs.

7. **Bias.** In terms of gender bias, minDALL-E, DALL-E mini, and Safe Stable Diffusion exhibit the least bias, while Dreamlike Diffusion, DALL-E 2, and Redshift Diffusion demonstrate higher levels of bias (https://crfm.stanford.edu/heim/v1.1.0/?group=heim_bias_scenarios). The mitigation of gender bias in Safe Stable Diffusion is intriguing, potentially due to its safety guidance mechanism suppressing sexual content. In terms of skin tone bias, Openjourney v2, CogView2, and GigaGAN show the least bias, whereas Dreamlike Diffusion and Redshift Diffusion exhibit more bias. Overall, minDALL-E consistently shows the least bias, while models fine-tuned on art images like Dreamlike and Redshift tend to exhibit more bias.

8. **Toxicity.** While most models exhibit a low frequency of generating inappropriate images, certain models exhibit a higher frequency for the I2P scenario (https://crfm.stanford.edu/heim/v1.1.0/?group=heim_toxicity_scenarios). For example, OpenJourney, the weaker variants of SafeStableDiffusion, Stable Diffusion, Promptist, and Vintedois Diffusion generate inappropriate images for non-toxic text prompts in over 10% of cases. The stronger variants of SafeStableDiffusion, which more strongly enforce safety guidance, generate fewer inappropriate images than Stable Diffusion but still produce inappropriate images. In contrast, models like minDALL-E, DALL-E mini, and GigaGAN exhibit the lowest frequency, less than 1%. This disparity may be attributed to the data used to train these models. Moving forward, addressing inappropriate image generation requires careful consideration of training data and model design.

9. **Fairness.** Around half of the models exhibit performance drops in human-rated alignment metrics when subjected to gender and dialect perturbations (https://crfm.stanford.edu/heim/v1.1.0/?group=mscoco_gender, https://crfm.stanford.edu/heim/v1.1.0/?group=mscoco_dialect). This suggests that there is no universal pattern of fairness issues among

the models. However, certain models incur significant performance drops, such as a -0.25 drop in human-rated alignment for Openjourney under dialect perturbation. In contrast, DALL-E mini showed the smallest performance gap in both scenarios. Overall, models fine-tuned on custom data displayed greater sensitivity to demographic perturbations.

10. **Robustness.** Similar to fairness, about half of the models showed performance drops in human-rated alignment metrics when typos were introduced (https://crfm.stanford.edu/heim/v1.1.0/?group=mscoco_robustness). These drops were generally minor, with the alignment score decreasing by no more than 0.2, indicating that these models are robust against prompt perturbations.

11. **Multilinguality.** Translating the MS-COCO prompts into Hindi, Chinese, and Spanish resulted in decreased image-text alignment for the vast majority of models (https://crfm.stanford.edu/heim/v1.1.0/?group=mscoco_chinese, https://crfm.stanford.edu/heim/v1.1.0/?group=mscoco_hindi, https://crfm.stanford.edu/heim/v1.1.0/?group=mscoco_spanish). A notable exception is CogView 2 for Chinese, which is known to perform better with Chinese prompts than with English prompts. DALL-E 2, the top model for human-rated image-text alignment (4.438 out of 5), maintains reasonable alignment with only a slight drop in performance for Chinese (-0.536) and Spanish (-0.162) prompts but struggles with Hindi prompts (-2.640). In general, the list of supported languages is not documented well for existing models, which motivates future practices to address this.

12. **Efficiency.** Among diffusion models, the vanilla Stable Diffusion has a denoised runtime of 2 seconds (https://crfm.stanford.edu/heim/v1.1.0/?group=heim_efficiency_scenarios). Methods with additional operations, such as prompt engineering in Promptist and safety guidance in SafeStableDiffusion, as well as models generating higher resolutions like Dreamlike Photoreal 2.0, exhibit slightly slower performance. Autoregressive models, like minDALL-E, are approximately 2 seconds slower than diffusion models with a similar parameter count. We also found that most models display significant variation in the raw runtime across multiple runs, possibly due to queuing of requests or interference among concurrent requests.

13. **Overall trends in aspects.** Among the current models, certain aspects exhibit positive correlations, such as general alignment and reasoning, as well as aesthetics and originality. On the other hand, some aspects show trade-offs; models excelling in aesthetics (e.g., Openjourney) tend to score lower in photorealism, and models proficient in bias and toxicity mitigation (e.g., minDALL-E) may not perform the best in image-text alignment and photorealism. Overall, several aspects deserve attention. Firstly, almost all models exhibit subpar performance in reasoning, photorealism, and multilinguality, highlighting the need for future improvements in these areas. Additionally, aspects like originality (watermarks), toxicity, and bias carry significant ethical and legal implications, yet current models are still imperfect and further research is necessary to address these concerns.

14. **Prompt engineering.** Models using prompt engineering techniques produce images that are more visually appealing. Promptist + Stable Diffusion v1-4 outperforms Stable Diffusion in terms of human-rated aesthetics score while achieving a comparable image-text alignment score (https://crfm.stanford.edu/heim/v1.1.0/?group=heim_quality_scenarios).

15. **Art styles.** According to human raters, Openjourney (fine-tuned on artistic images generated by Midjourney) creates the most aesthetically pleasing images across the various art styles (https://crfm.stanford.edu/heim/v1.1.0/?group=mscoco_art_styles). It is followed by Dreamlike Photoreal 2.0 and DALL-E 2. DALL-E 2 achieves the highest human-rated alignment score. Dreamlike Photoreal 2.0 (Stable Diffusion fine-tuned on high-resolution photographs) demonstrates superior human-rated subject clarity.

16. **Correlation between human and automated metrics.** The correlation coefficients between human-rated and automated metrics are 0.42 for alignment (CLIPScore vs human-rated alignment), 0.59 for image quality (FID vs human-rated photorealism), and 0.39 for aesthetics (LAION aesthetics vs human-rated aesthetics) (https://crfm.stanford.edu/heim/v1.1.0/?group=mscoco_fid, https://crfm.stanford.edu/heim/v1.1.0/?group=mscoco_base). The overall correlation is weak, particularly for aesthetics. These findings emphasize the importance of using human ratings for evaluating image generation models in future research.

17. **Diffusion vs autoregressive models.** Among the open-sourced autoregressive and diffusion models, autoregressive models require a larger model size to achieve performance comparable to diffusion models across most metrics. Nevertheless, autoregressive models show promising performance in some aspects, such as reasoning. Diffusion models exhibit greater efficiency compared to autoregressive models when controlling for parameter count.

18. **Model scales.** Multiple models with varying parameter counts are available within the autoregressive DALL-E model family (0.4B, 1.3B, 2.6B) and diffusion DeepFloyd-IF family (0.4B, 0.9B, 4.3B). We find that larger models tend to outperform smaller ones in all human-rated metrics, including alignment, photorealism, subject clarity, and aesthetics (https://crfm.stanford.edu/heim/v1.1.0/?group=mscoco_base).

19. **What are the best models?** Overall, DALL-E 2 appears to be a versatile performer across human-rated metrics. However, no single model emerges as the top performer in all aspects. Different models show different strengths. For example, Dreamlike Photoreal excel in photorealism, while Openjourney in aesthetics. For societal aspects, models like minDALL-E, CogView2, and SafeStableDiffusion perform well in toxicity and bias mitigation. For multilinguality, GigaGAN and the DeepFloyd-IF models seem to handle Hindi prompts, which DALL-E 2 struggles with. These observations open up new research avenues to study whether and how to develop models that excel across multiple aspects.

## 8 Related work

**Holistic benchmarking.** Benchmarks drive the advancements of AI by orienting the directions for the community to improve upon [20, 54, 55, 56]. In particular, in natural language processing (NLP), the adoption of meta-benchmarks [57, 58, 59, 60] and holistic evaluation [1] across multiple scenarios or tasks has allowed for comprehensive assessments of models and accelerated model improvements. However, despite the growing popularity of image generation and the increasing number of models being developed, a holistic evaluation of these models has been lacking. Furthermore, image generation encompasses various technological and societal impacts, including alignment, quality, originality, toxicity, bias, and fairness, which necessitate comprehensive evaluation. Our work fills this gap by conducting a holistic evaluation of image generation models across 12 important aspects.

**Benchmarks for image generation.** Existing benchmarks primarily focus on assessing image quality and alignment, using automated metrics. Widely used benchmarks such as MS-COCO [21] and ImageNet [20] have been employed to evaluate the quality and alignment of generated images. Metrics like Fréchet Inception Distance (FID) [23], Inception Score [36], and CLIPScore [24] are commonly used for quantitative assessment of image quality and alignment.

To better capture human perception in image evaluation, crowdsourced human evaluation has been explored in recent years [25, 6, 35, 61]. However, these evaluations have been limited to assessing aspects such as alignment and quality. Building upon these crowdsourcing techniques, we extend the evaluation to include additional aspects such as aesthetics, originality, reasoning, and fairness.

As the ethical and societal impacts of image generation models gain prominence [19], researchers have also started evaluating these aspects [33, 29, 8]. However, these evaluations have been conducted on only a select few models, leaving the majority of models unevaluated in these aspects. Our standardized evaluation addresses this gap by enabling the evaluation of all models across all aspects, including ethical and societal dimensions.

**Art and design.** Our assessment of image generation incorporates aesthetic evaluation and design principles. Aesthetic evaluation considers factors like composition, color harmony, balance, and visual complexity [62, 63]. Design principles, such as clarity, legibility, hierarchy, and consistency in design elements, also influence our evaluation [64]. Combining these insights, our aim is to determine whether generated images are visually pleasing, with thoughtful compositions, harmonious colors, balanced elements, and an appropriate level of visual complexity. We employ objective metrics and subjective human ratings for a comprehensive assessment of aesthetic quality.

## 9 Conclusion

We introduced Holistic Evaluation of Text-to-Image Models (HEIM), a new benchmark to assess 12 important aspects in text-to-image generation, including alignment, quality, aesthetics, originality, reasoning, knowledge, bias, toxicity, fairness, robustness, multilinguality, and efficiency. Our evaluation of 26 recent text-to-image models reveals that different models excel in different aspects, opening up research avenues to study whether and how to develop models that excel across multiple aspects. To enhance transparency and reproducibility, we release our evaluation pipeline, along with the generated images and human evaluation results. We encourage the community to consider the different aspects when developing text-to-image models.

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

## Acknowledgments

We thank Robin Rombach, Yuhui Zhang, members of Stanford P-Lambda, CRFM, and SNAP groups, as well as our anonymous reviewers for providing valuable feedback. We thank Josselin Somerville for assisting with the human evaluation infrastructure. This work is supported in part by the AI2050 program at Schmidt Futures (Grant G-22-63429) and ONR N00014-23-1-2355. Michihiro Yasunaga is supported by a Microsoft Research PhD Fellowship.

## A    Author contributions

**Tony Lee**: Led the project; Designed the core framework (aspects, scenarios, metrics); Implemented scenarios, metrics and models; Conducted experiments. Contributed to writing.

**Michihiro Yasunaga**: Led the project; Designed the core framework (aspects, scenarios, metrics); Wrote the paper; Conducted analysis; Implemented models.

**Chenlin Meng**: Designed the core framework (aspects, scenarios, metrics). Contributed to writing.

**Yifan Mai**: Implemented the evaluation infrastructure; Contributed to project discussions

**Joon Sung Park**: Designed human evaluation questions

**Agrim Gupta**: Implemented the detection scenario and metrics

**Yunzhi Zhang**: Implemented the detection scenario and metrics

**Deepak Narayanan**: Provided expertise and analysis of efficiency metrics

**Hannah Teufel**: Provided model expertise and inference

**Marco Bellagente**: Provided model expertise and inference

**Minguk Kang**: Provided model expertise and inference

**Taesung Park**: Provided model expertise and inference

**Jure Leskovec**: Provided advice on human evaluation and paper writing

**Jun-Yan Zhu**: Provided advice on human evaluation

**Li Fei-Fei**: Provided advice on the core framework

**Jiajun Wu**: Provided advice on the core framework

**Stefano Ermon**: Provided advice on the core framework

**Percy Liang**: Provided overall supervision and guidance throughout the project

### A.1    Statement of neutrality

The authors of this paper affirm their commitment to maintaining a fair and independent evaluation of the image generation models. We acknowledge that the author affiliations encompass a range of academic and industrial institutions, including those where some of the models we evaluate were developed. However, the authors' involvement is solely based on their expertise and efforts to run and evaluate the models, and the authors have treated all models equally throughout the evaluation process, regardless of their sources. This study aims to provide an objective understanding and assessment of models across various aspects, and we do not intend to endorse specific models.

## B    Limitations

Our work identifies 12 important aspects in real-world deployments of text-to-image generation models, namely alignment, quality, aesthetics, originality, reasoning, knowledge, bias, toxicity, fairness, robustness, multilinguality, and efficiency. While we have made substantial progress in conducting a holistic evaluation of models across these aspects, there are certain limitations that should be acknowledged in our work.

Firstly, it is important to note that our identified 12 aspects may not be exhaustive, and there could be other potentially important aspects in text-to-image generation that have not been considered. It is an ongoing area of research, and future studies may uncover additional dimensions that are critical for evaluating image generation models. We welcome further exploration in this direction to ensure a comprehensive understanding of the field.

Secondly, our current metrics for evaluating certain aspects may not be exhaustive. For instance, when assessing bias, our current focus lies on binary gender and skin tone representations, yet there may be other demographic factors that warrant consideration. Additionally, our assessment of efficiency currently relies on measuring wall-clock time, which directly captures latency but merely acts as a surrogate for the actual energy consumption of the models. In our future work, we intend to expand our metrics to enable a more comprehensive evaluation of each aspect.

Lastly, there is an additional limitation related to the use of crowdsourced human evaluation. While crowdsource workers can effectively answer certain evaluation questions, such as image alignment, photorealism, and subject clarity, and provide a high level of inter-annotator agreement, there are other aspects, namely overall aesthetics and originality, where the responses from crowdsource workers (representing the general public) may exhibit greater variance. These metrics rely on subjective judgments, and it is acknowledged that the opinions of professional artists or legal experts may differ from those of the general public. Consequently, we refrain from drawing strong conclusions based solely on these metrics. However, we do believe there is value in considering the judgments of the general public, as it is reasonable to desire generated images to be visually pleasing and exhibit a sense of originality to a wide audience.

## C   Datasheet

### C.1   Motivation

Q1 **For what purpose was the dataset created?** Was there a specific task in mind? Was there a specific gap that needed to be filled? Please provide a description.

- The HEIM benchmark was created to holistically evaluate text-to-image models across diverse aspects. Before HEIM, text-to-image models were typically evaluated on the alignment and quality aspects; with HEIM, we evaluate across 12 different aspects that are important in real-world model deployment: image alignment, quality, aesthetics, originality, reasoning, knowledge, bias, toxicity, fairness, robustness, multilinguality and efficiency.

Q2 **Who created the dataset (e.g., which team, research group) and on behalf of which entity (e.g., company, institution, organization)?**

- This benchmark is presented by the Center for Research on Foundation Models (CRFM), an interdisciplinary initiative born out of the Stanford Institute for Human-Centered Artificial Intelligence (HAI) that aims to make fundamental advances in the study, development, and deployment of foundation models. `https://crfm.stanford.edu/`.

Q3 **Who funded the creation of the dataset?** If there is an associated grant, please provide the name of the grantor and the grant name and number.

- This work was supported in part by the AI2050 program at Schmidt Futures (Grant G-22-63429).

Q4 **Any other comments?**

- No.

### C.2   Composition

Q5 **What do the instances that comprise the dataset represent (e.g., documents, photos, people, countries)?** *Are there multiple types of instances (e.g., movies, users, and ratings; people and interactions between them; nodes and edges)? Please provide a description.*

- HEIM benchmark provides prompts/captions covering 62 scenarios. We also release images generated by 26 text-to-image models from these prompts.

Q6 **How many instances are there in total (of each type, if appropriate)?**

- HEIM contains 500K prompts in total covering 62 scenarios. The detailed statistics for each scenario can be found at `https://crfm.stanford.edu/heim/latest`.

Q7 **Does the dataset contain all possible instances or is it a sample (not necessarily random) of instances from a larger set?** *If the dataset is a sample, then what is the larger set? Is the sample representative of the larger set (e.g., geographic coverage)? If so, please describe how this representativeness was validated/verified. If it is not representative of the larger set, please describe why not (e.g., to cover a more diverse range of instances, because instances were withheld or unavailable).*

- Yes. The scenarios in our benchmark are sourced by existing datasets such as MS-COCO, DrawBench, PartiPrompts, etc. and we use all possible instances from these datasets.

Q8 **What data does each instance consist of?** *"Raw" data (e.g., unprocessed text or images) or features? In either case, please provide a description.*

- Input prompt and generated images.

Q9 **Is there a label or target associated with each instance?** *If so, please provide a description.*

- The MS-COCO scenario contains a reference image for every prompt, as in the original MS-COCO dataset. Other scenarios do not have reference images.

Q10 **Is any information missing from individual instances?** *If so, please provide a description, explaining why this information is missing (e.g., because it was unavailable). This does not include intentionally removed information, but might include, e.g., redacted text.*

- No.

Q11 **Are relationships between individual instances made explicit (e.g., users' movie ratings, social network links)?** *If so, please describe how these relationships are made explicit.*

- Every prompt belongs to a scenario.

Q12 **Are there recommended data splits (e.g., training, development/validation, testing)?** *If so, please provide a description of these splits, explaining the rationale behind them.*

- No.

Q13 **Are there any errors, sources of noise, or redundancies in the dataset?** *If so, please provide a description.*

- No.

Q14 **Is the dataset self-contained, or does it link to or otherwise rely on external resources (e.g., websites, tweets, other datasets)?** *If it links to or relies on external resources, a) are there guarantees that they will exist, and remain constant, over time; b) are there official archival versions of the complete dataset (i.e., including the external resources as they existed at the time the dataset was created); c) are there any restrictions (e.g., licenses, fees) associated with any of the external resources that might apply to a future user? Please provide descriptions of all external resources and any restrictions associated with them, as well as links or other access points, as appropriate.*

- The dataset is self-contained. Everything is available at https://crfm.stanford.edu/heim/latest.

Q15 **Does the dataset contain data that might be considered confidential (e.g., data that is protected by legal privilege or by doctor–patient confidentiality, data that includes the content of individuals' non-public communications)?** *If so, please provide a description.*

- No. The majority of scenarios used in our benchmark are sourced from existing open-source datasets. The new scenarios we introduced in this work, namely Historical Figures, DailyDall.e, Landing Pages, Logos, Magazine Covers, and Mental Disorders, were also constructed by using public resources.

Q16 **Does the dataset contain data that, if viewed directly, might be offensive, insulting, threatening, or might otherwise cause anxiety?** *If so, please describe why.*

- We release all images generated by models, which may contain sexually explicit, racist, abusive or other discomforting or disturbing content. For scenarios that are likely to contain such inappropriate images, our website will display this warning. We release all images with the hope that they can be useful for future research studying the safety of image generation outputs.

Q17 **Does the dataset relate to people?** *If not, you may skip the remaining questions in this section.*

- People may be present in the prompts or generated images, but people are not the sole focus of the dataset.

Q18 **Does the dataset identify any subpopulations (e.g., by age, gender)?**

- We use automated gender and skin tone classifiers for evaluating biases in generated images.

Q19 **Is it possible to identify individuals (i.e., one or more natural persons), either directly or indirectly (i.e., in combination with other data) from the dataset?** *If so, please describe how.*

- Yes it may be possible to identify people in the generated images using face recognition. Similarly, people may be identified through the associated prompts.

Q20 **Does the dataset contain data that might be considered sensitive in any way (e.g., data that reveals racial or ethnic origins, sexual orientations, religious beliefs, political opinions or union memberships, or locations; financial or health data; biometric or genetic data; forms of government identification, such as social security numbers; criminal history)?** *If so, please provide a description.*

- Yes the model-generated images contain sensitive content. The goal of our work is to evaluate the toxicity and bias in these generated images.

Q21 **Any other comments?**

- We caution discretion on behalf of the user and call for responsible usage of the benchmark for research purposes only.

### C.3 Collection Process

Q22 **How was the data associated with each instance acquired?** *Was the data directly observable (e.g., raw text, movie ratings), reported by subjects (e.g., survey responses), or indirectly inferred/derived from other data (e.g., part-of-speech tags, model-based guesses for age or language)? If data was reported by subjects or indirectly inferred/derived from other data, was the data validated/verified? If so, please describe how.*

- The majority of scenarios used in our benchmark are sourced from existing open-source datasets, which are referenced in Table 2.
- For the new scenarios we introduce in this work, namely Historical Figures, DailyDall.e, Landing Pages, Logos, Magazine Covers, and Mental Disorders, we collected or wrote the prompts.

Q23 **What mechanisms or procedures were used to collect the data (e.g., hardware apparatus or sensor, manual human curation, software program, software API)?** *How were these mechanisms or procedures validated?*

- The existing scenarios were downloaded by us.
- Prompts for the new scenarios were collected or written by us manually. For further details, please refer to §D.

Q24 **If the dataset is a sample from a larger set, what was the sampling strategy (e.g., deterministic, probabilistic with specific sampling probabilities)?**

- We use the whole datasets

Q25 **Who was involved in the data collection process (e.g., students, crowdworkers, contractors) and how were they compensated (e.g., how much were crowdworkers paid)?**

- The authors of this paper collected the scenarios.
- Crowdworkers were only involved when we evaluate images generated by models from these scenarios.

Q26 **Over what timeframe was the data collected? Does this timeframe match the creation timeframe of the data associated with the instances (e.g., recent crawl of old news articles)?** *If not, please describe the timeframe in which the data associated with the instances was created.*

- The data was collected from December 2022 to June 2023.

Q27 **Were any ethical review processes conducted (e.g., by an institutional review board)?** *If so, please provide a description of these review processes, including the outcomes, as well as a link or other access point to any supporting documentation.*

- We corresponded with the Research Compliance Office at Stanford University. After submitting an application with our research proposal and details of the human evaluation, Adam Bailey, the Social and Behavior (non-medical) Senior IRB Manager, deemed that our IRB protocol 69233 did not meet the regulatory definition of human subjects research since we do not plan to draw conclusions about humans nor are we evaluating any characteristics of the human raters. As such, the protocol has been withdrawn, and we were allowed to work on this research project without any additional IRB review.

Q28 **Does the dataset relate to people?** *If not, you may skip the remaining questions in this section.*

- People may appear in the images and descriptions, although they are not the exclusive focus of the dataset.

Q29 **Did you collect the data from the individuals in question directly, or obtain it via third parties or other sources (e.g., websites)?**

- Our scenarios were collected from third party websites. Our human evaluation were conducted via crowdsourcing.

Q30 **Were the individuals in question notified about the data collection?** *If so, please describe (or show with screenshots or other information) how notice was provided, and provide a link or other access point to, or otherwise reproduce, the exact language of the notification itself.*

- Individuals involved in crowdsourced human evaluation were notified about the data collection. We used Amazon Mechanical Turk, and presented the consent form shown in Figure 5 to the crowdsource workers.

DESCRIPTION: You are invited to participate in a research study on evaluating text-to-image generation models. You will rate A.I.-generated images based on criteria such as image quality, creativity, aesthetics, etc.

TIME INVOLVEMENT: Your participation will take approximately 20 minutes.

RISK AND BENEFITS: The risks associated with this study are that some of the images in this study can be inappropriate or toxic - images can contain violence, threats, obscenity, insults, profanity, or sexually explicit content. Study data will be stored securely, in compliance with Stanford University standards, minimizing the risk of a confidentiality breach. There are no immediate non-monetary benefits from this study. We cannot and do not guarantee or promise that you will receive any benefits from this study. AI image generation models are becoming increasingly pervasive in society. As a result, understanding their characteristics has become important for the benefit of society.

PARTICIPANT'S RIGHTS: If you have read this form and have decided to participate in this project, please understand your participation is voluntary, and you have the right to withdraw your consent or discontinue participation at any time without penalty or loss of benefits to which you are otherwise entitled. The alternative is not to participate. You have the right to refuse to answer particular questions. The results of this research study may be presented at scientific or professional meetings or published in scientific journals. Your individual privacy will be maintained in all published and written data resulting from the study.

CONTACT INFORMATION: Questions: If you have any questions, concerns, or complaints about this research, its procedures, risks, and benefits, contact the Protocol Director, Percy Liang, at (650) 723-6319.

Independent Contact: If you are not satisfied with how this study is being conducted, or if you have any concerns, complaints, or general questions about the research or your rights as a participant, please contact the Stanford Institutional Review Board (IRB) to speak to someone independent of the research team at 650-723-2480 or toll-free at 1-866-680-2906 or email at irbnonmed@stanford.edu. You can also write to the Stanford IRB, Stanford University, 1705 El Camino Real, Palo Alto, CA 94306.

Please save or print a copy of this page for your records.

If you agree to participate in this research, please start the session.

Figure 5: Consent form for data collection, used in the crowdsourced human evaluation

Q31 **Did the individuals in question consent to the collection and use of their data?** *If so, please describe (or show with screenshots or other information) how consent was requested and provided, and provide a link or other access point to, or otherwise reproduce, the exact language to which the individuals consented.*

- Yes, consent was obtained from crowdsource workers for human evaluation. Please refer to Figure 5.

Q32 **If consent was obtained, were the consenting individuals provided with a mechanism to revoke their consent in the future or for certain uses?** *If so, please provide a description, as well as a link or other access point to the mechanism (if appropriate).*

- Yes. Please refer to Figure 5.

Q33 **Has an analysis of the potential impact of the dataset and its use on data subjects (e.g., a data protection impact analysis) been conducted?** *If so, please provide a description of this analysis, including the outcomes, as well as a link or other access point to any supporting documentation.*

- We discuss the limitation of our current work in §B, and we plan to further investigate and analyze the impact of our benchmark in future work.

Q34 **Any other comments?**

- No.

## C.4 Preprocessing, Cleaning, and/or Labeling

Q35 **Was any preprocessing/cleaning/labeling of the data done (e.g., discretization or bucketing, tokenization, part-of-speech tagging, SIFT feature extraction, removal of instances, processing of missing values)?** *If so, please provide a description. If not, you may skip the remainder of the questions in this section.*

- No preprocessing or labelling was done for creating the scenarios.

Q36 **Was the "raw" data saved in addition to the preprocessed/cleaned/labeled data (e.g., to support unanticipated future uses)?** *If so, please provide a link or other access point to the "raw" data.*

- N/A. No preprocessing or labelling was done for creating the scenarios.

Q37 **Is the software used to preprocess/clean/label the instances available?** *If so, please provide a link or other access point.*

- N/A. No preprocessing or labelling was done for creating the scenarios.

Q38 **Any other comments?**

- No.

## C.5 Uses

Q39 **Has the dataset been used for any tasks already?** *If so, please provide a description.*

- Not yet. HEIM is a new benchmark.

Q40 **Is there a repository that links to any or all papers or systems that use the dataset?** *If so, please provide a link or other access point.*

- We will provide links to works that use our benchmark at https://crfm.stanford.edu/heim/latest.

Q41 **What (other) tasks could the dataset be used for?**

- The primary use case of our benchmark is text-to-image generation.
- While we did not explore this direction in the present work, the prompt-image pairs available in our benchmark may be used for image-to-text generation research in future.

Q42 **Is there anything about the composition of the dataset or the way it was collected and preprocessed/cleaned/labeled that might impact future uses?** *For example, is there anything that a future user might need to know to avoid uses that could result in unfair treatment of individuals or groups (e.g., stereotyping, quality of service issues) or other undesirable harms (e.g., financial harms, legal risks) If so, please provide a description. Is there anything a future user could do to mitigate these undesirable harms?*

- Our benchmark contains images generated by models, which may exhibit biases in demographics and contain toxic contents such as violence and nudity. The images released by this benchmark should not be used to make a decision surrounding people.

Q43 **Are there tasks for which the dataset should not be used?** *If so, please provide a description.*

- Because the model-generated images in this benchmark may contain bias and toxic content, under no circumstance should these images or models trained on them be put into production. It is neither safe nor responsible. As it stands, the images should be solely used for research purposes.
- Likewise, this benchmark should not be used to aid in military or surveillance tasks.

Q44 **Any other comments?**

- No.

## C.6 Distribution and License

Q45 **Will the dataset be distributed to third parties outside of the entity (e.g., company, institution, organization) on behalf of which the dataset was created?** *If so, please provide a description.*

- Yes, this benchmark will be open-source.

Q46 **How will the dataset be distributed (e.g., tarball on website, API, GitHub)?** *Does the dataset have a digital object identifier (DOI)?*

- Our data (scenarios, generated images, evaluation results) are available at `https://crfm.stanford.edu/heim/latest`.
- Our code used for evaluation is available at `https://github.com/stanford-crfm/helm`.

**Q47  When will the dataset be distributed?**

- June 7, 2023 and onward.

**Q48  Will the dataset be distributed under a copyright or other intellectual property (IP) license, and/or under applicable terms of use (ToU)?** *If so, please describe this license and/or ToU, and provide a link or other access point to, or otherwise reproduce, any relevant licensing terms or ToU, as well as any fees associated with these restrictions.*

- The majority of scenarios used in our benchmark are sourced from existing open-source datasets, which are referenced in Table 2. The license associated with them is followed accordingly.
- We release the new scenarios, namely Historical Figures, DailyDall.e, Landing Pages, Logos, Magazine Covers, and Mental Disorders, under the **CC-BY-4.0** license.
- Our code is released under the **Apache-2.0** license

**Q49  Have any third parties imposed IP-based or other restrictions on the data associated with the instances?** *If so, please describe these restrictions, and provide a link or other access point to, or otherwise reproduce, any relevant licensing terms, as well as any fees associated with these restrictions.*

- We own the metadata and release as CC-BY-4.0.
- We do not own the copyright of the images or text.

**Q50  Do any export controls or other regulatory restrictions apply to the dataset or to individual instances?** *If so, please describe these restrictions, and provide a link or other access point to, or otherwise reproduce, any supporting documentation.*

- No.

**Q51  Any other comments?**

- No.

### C.7   Maintenance

**Q52  Who will be supporting/hosting/maintaining the dataset?**

- Stanford CRFM will be supporting, hosting, and maintaining the benchmark.

**Q53  How can the owner/curator/manager of the dataset be contacted (e.g., email address)?**

- `https://crfm.stanford.edu`

**Q54  Is there an erratum?** *If so, please provide a link or other access point.*

- There is no erratum for our initial release. Errata will be documented as future releases on the benchmark website.

**Q55  Will the dataset be updated (e.g., to correct labeling errors, add new instances, delete instances)?** *If so, please describe how often, by whom, and how updates will be communicated to users (e.g., mailing list, GitHub)?*

- HEIM will be updated. We plan to expand scenarios, metrics, and models to be evaluated.

**Q56  If the dataset relates to people, are there applicable limits on the retention of the data associated with the instances (e.g., were individuals in question told that their data would be retained for a fixed period of time and then deleted)?** *If so, please describe these limits and explain how they will be enforced.*

- People may contact us at `https://crfm.stanford.edu` to add specific samples to a blacklist.

 **Will older versions of the dataset continue to be supported/hosted/maintained?** *If so, please describe how. If not, please describe how its obsolescence will be communicated to users.*

- We will host other versions.

Q58 **If others want to extend/augment/build on/contribute to the dataset, is there a mechanism for them to do so?** *If so, please provide a description. Will these contributions be validated/verified? If so, please describe how. If not, why not? Is there a process for communicating/distributing these contributions to other users? If so, please provide a description.*

- People may contact us at `https://crfm.stanford.edu` to request adding new scenarios, metrics, or models.

Q59 **Any other comments?**

- No.

# D  Scenario details

## D.1  Existing scenarios

**MS-COCO.**   MS COCO [21] is a large-scale labeled image dataset containing images of humans and everyday objects. Examples of the caption include "A large bus sitting next to a very tall building", "The man at bad readies to swing at the pitch while the umpire looks on", "Bunk bed with a narrow shelf sitting underneath it". We use the 2014 validation set (40,504 captions) to generate images for evaluating image *quality*, text-image *alignment*, and inference *efficiency*.

**CUB-200-2011.**   CUB-200-2011 [65] is a challenging paired text-image dataset of 200 bird species. It contains 29,930 captions. Example captions include: "Acadian flycatcher", "American goldfinch", "Cape May warbler". We use captions from the dataset for evaluating the image-text *alignment* of the models.

**DrawBench.**   DrawBench [6] is a structured suite of 200 text prompts designed for probing the semantic properties of text-to-image models. These properties include compositionality, cardinality, spatial relations, and many more. Example text prompts include "A black apple and a green backpack" (Colors), "Three cats and one dog sitting on the grass" (Counting), "A stop sign on the right of a refrigerator" (Positional). We use text prompts from DrawBench for evaluating the *alignment*, *quality*, *reasoning* and *knowledge* aspects of the text-to-image models.

**PartiPrompts.**   PartiPrompts (P2) [7] is a benchmark dataset consisting of over 1600 English prompts. It includes categories such as Artifacts, Food & Beverage, Vehicles, Arts, Indoor Scenes, Outdoor Scenes, Produce & Plants, People, Animals, Illustrations. Example text prompts include "A portrait photo of a kangaroo wearing an orange hoodie and blue sunglasses standing on the grass in front of the Sydney Opera House holding a sign on the chest that says Welcome Friends!", "A green sign that says "Very Deep Learning" and is at the edge of the Grand Canyon. Puffy white clouds are in the sky", "A photo of an astronaut riding a horse in the forest. There is a river in front of them with water lilies". We use text prompts from P2 for evaluating the text-image *alignment*, *reasoning* and *knowledge* aspects of the models.

**Relational Understanding** [31]**.**   This scenario aims to assess the reasoning abilities of text-to-image models. Drawing from cognitive, linguistic, and developmental literature, a collection of 15 relations (8 physical and 7 agentic) and 12 entities (6 objects and 6 agents) has been compiled. A total of 75 prompts have been constructed, involving the combination of these objects and relations. Examples of prompts include "a man pushing a box" and "a robot pulling a monkey".

**Detection** [29]**.**   To assess the visual reasoning capabilities of text-to-image models, we adopt the prompt templates used in [29] and evaluate three specific skills: object recognition, object counting, and spatial relation understanding. For evaluation, we use the 15 most common MS-COCO [21] classes and four spatial relations: ABOVE, BELOW, LEFT, and RIGHT. This results in a total of 21 prompts for object recognition, 84 prompts for object counting, and 1764 prompts for spatial relation understanding. Examples of prompts include "a photo of 3 suitcases" (counting) and "A potted plant is below a bird" (spatial relation).

**Winoground** [32].    This scenario focuses on examining the reasoning capabilities of text-to-image models. It contains 400 prompts that require precise understanding of word orders, "some plants surrounding a lightbulb" vs " a lightbulb surrounding some plants".

**Common Syntactic Processes** [30].    This scenario focuses on examining the reasoning capabilities of text-to-image models. Prompts capture 8 grammatical phenomena pertaining to compositionality that are widely discussed in linguistics and pervasive in human language: binding principles and coreference (10 prompts), passives (6), word order (10), coordination (5), comparatives (5), negation (5), ellipsis (5), and structural ambiguity (6). Examples of prompts include "A tall woman without a handbag" (negation) and "The bowl has more cucumbers than strawberries" (comparatives).

**Demographic Stereotypes** [33, 29].    This scenario focuses on examining the presence of demographic bias in text-to-image generation outputs. The prompts in this scenario involve descriptors (15 prompts) or occupations (13 prompts) of people, some of which may exhibit stereotypical associations with demographic groups. Examples of the prompts include "a photo of the face of a poor person" and "a photo of the face of a chef".

**Inappropriate Image Prompts (I2P)** [8].    This scenario focuses on examining whether models generate inappropriate images. Collects real user prompts that are likely to produce such images. It has 7 categories of inappropriate content: Hate (89 prompts), Harassment (261), Violence (310), Self-harm (309), Sexual content (296), Shocking image (466), Illegal activity (236). An example of the prompts include "a photography of a person in a bath tub".

### D.2    New scenarios

**MS-COCO (art).**    Artists and users of text-to-image models often desire images in specific art styles (e.g., "dogs playing poker, Baroque period art"). To test the ability of these models to generate images in specific art styles, we created an art version of the MS-COCO scenario. We take the original MS-COCO prompts and append the name corresponding to the art style: "[original prompt], [art style]". We used the following common art styles for this scenario: *oil painting*, *watercolor*, *pencil sketch*, *animation*, *vector graphics*, and *pixel art*. An example prompt is "A large bus sitting next to a very tall building, pencil sketch" where the art style "pencil sketch" was appended to the original prompt. This scenario is used to evaluate the models on the *aesthetics* aspect.

**MS-COCO (fairness – gender).**    Following [1], we measure the fairness of using male vs. gender terms. We take the original prompts from MS-COCO and map male gender terms to female gender terms (e.g., "son" to "daughter" and "father" to "mother"). An example of this transformation for MS-COCO is "People staring at a *man* on a fancy motorcycle." is updated to "People staring at a *woman* on a fancy motorcycle."

**MS-COCO (fairness – African-American English dialect).**    Going from Standard American English to African American English for the GLUE benchmark can lead to a drop in model performance [66]. Following what was done for language models in [1], we measure the fairness for the speaker property of Standard American English vs. African American English for text-to-image models. We take the original prompts from MS-COCO and convert each word to the corresponding word in African American Vernacular English if one exists. For example, the prompt "A birthday cake explicit in nature makes a *girl* laugh." is transformed to "A birthday cake explicit in nature makes a *gurl* laugh."

**MS-COCO (robustness – typos).**    Similar to how [1] measured how robust language models are to invariant perturbations, we modify the MS-COCO prompts in a semantic-preserving manner by following these steps:

1. Lowercase all letters.

2. Replace each expansion with its contracted version (e.g., "She is a doctor, and I am a student" to "She's a doctor, and I'm a student").

3. Replace each word with a common misspelling with 0.1 probability.

4. Replace each whitespace with 1, 2, or 3 whitespaces.

For example, the prompt "A horse standing in a field that is genetically part zebra." is transformed to "a horse standing in a field that's genetically part zebra.", preserving the original meaning of the sentence.

**MS-COCO (languages).**    In order to reach a wider audience, it is critical for AI systems to support multiple languages besides English. Therefore, we translate the MS-COCO prompts from English to the three most commonly spoken languages using Google's Cloud Translation API: Chinese, Spanish, and Hindi. For example, the prompt "A man is serving grilled hot dogs in buns." is translated to:

- Chinese: 一个男人正在端着包在面包里的烤热狗。
- Hindi: एक आदमी बन्स में ग्रिल्ड हॉट डॉग परोस रहा है।
- Spanish: Un hombre está sirviendo perritos calientes a la parrilla en panecillos.

**Historical Figures.**    Historical figures serve as suitable entities to assess the knowledge of text-to-image models. For this purpose, we have curated 99 prompts following the format of "X", where X represents the name of a historical figure (e.g., "Napoleon Bonaparte"). The list of historical figures is sourced from TIME's compilation of The 100 Most Significant Figures in History: https://ideas.time.com/2013/12/10/whos-biggest-the-100-most-significant-figures-in-history.

**Dailydall.e.**    Chad Nelson is an artist who shares prompts on his Instagram account (https://www.instagram.com/dailydall.e), which he utilizes for generating artistic images using text-to-image models like DALL-E 2. To ensure our benchmark includes scenarios that are relevant and meaningful to artists, we have gathered 93 prompts from Chad Nelson's Instagram. For instance, one of the prompts reads, "close-up of a snow leopard in the snow hunting, rack focus, nature photography." This scenario can be used to assess the aesthetics and originality aspects.

**Landing Pages.**    A landing page is a single web page designed for a specific marketing or promotional purpose, typically aiming to convert visitors into customers or leads. Image generation models can potentially aid in creating landing pages by generating visually appealing elements such as images, illustrations, or layouts, enhancing the overall design and user experience. We have created 36 prompts for generating landing pages, following the format "a landing page of X" where X is a description of a website (e.g., "finance web application"). This scenario can be used to assess the aesthetics and originality aspects.

**Logos.**    A logo is a unique visual symbol or mark that represents a brand, company, or organization. It helps establish brand identity, build recognition, and convey the values and essence of the entity. Image generation models have the potential to assist in logo design by generating innovative logo concepts and offering a wider range of creative options. We have created 100 prompts for generating logos, following the format "a logo of X that does Y", where X is a type of the entity (e.g., "technology company") and Y is a specific description of the entity (e.g., "makes smartphones and personal computers"). This scenario can be used to assess the aesthetics and originality aspects.

**Magazine Covers.**    A magazine cover is the front page that visually represents a magazine's content and brand, aiming to attract readers and convey the magazine's essence. Image generation models can potentially generate visually appealing designs efficiently and offer creative possibilities for customization. We have created 50 prompts for generating magazine cover photos, following the format "a photo for a cover of a magazine with the headline: X", where X is a headline (e.g., "Bright ideas: Our favorite ways to make Thanksgiving sparkle"). The headlines are from the following magazines: *Better Homes & Gardens*, *Cosmopolitan*, *Costco Connection*, *National Geographic*, *Parents*, *Sports Illustrated*, and *TIME*. This scenario can be used to assess the aesthetics and originality aspects.

**Mental Disorders.**    One in every eight people in the world lives with a mental disorder. It is crucial for us to avoid harboring harmful biases regarding these disorders or the individuals who experience them. With inspiration from [34], we aim to evaluate the stereotypical associations present in text-to-image models regarding mental disorders. For this purpose, we have created nine prompts following the format "A photo of the face of a person with X," where X represents a mental disorder (e.g., "depression"). The list of mental disorders is sourced from the World Health Organization (WHO) website at https:

//www.who.int/news-room/fact-sheets/detail/mental-disorders. This scenario can be used to assess the bias aspect.

# E   Metric details

## E.1   Human-rated metrics

We rely on human annotators to evaluate the generated images based on several aspects: alignment, quality, aesthetics, and originality. For quality, we focus on photorealism. For aesthetics, we focus on subject clarity and overall aesthetics of the generated images. The following is the full detail of the human evaluation questions.

To obtain reliable human evaluation results, we employ crowdsourcing methodology in [35]. Concrete word descriptions are provided for each question and rating choice, and a minimum of 5 crowdsource workers evaluate each image. We use at least 100 image samples for each aspect being evaluated. For a more detailed description of the crowdsourcing procedure, see §G.

**Overall alignment.**    We investigate whether the generated image meets the annotators' expectations by asking them to rate how well the image matches the description using a 5-point Likert scale, similar to [35]:

How well does the image match the description?
a) Does not match at all
b) Has significant discrepancies
c) Has several minor discrepancies
d) Has a few minor discrepancies
e) Matches exactly

**Photorealism.**    While photorealism alone does not guarantee superior quality in all contexts, we include it as a measure to assess the basic competence of the text-to-image model. To evaluate photorealism, we employ the $\text{HYPE}_\infty$ metric [25], where annotators distinguish between real and model-generated images based on 200 samples, with 100 being real and 100 being model-generated. Following [35], below is the multiple-choice question asked of human annotators for both real and generated images:

Determine if the following image is AI-generated or real.
a) AI-generated photo
b) Probably an AI-generated photo, but photorealistic
c) Neutral
d) Probably a real photo, but with irregular textures and shapes
e) Real photo

**Subject clarity.**    We assess the subject clarity by evaluating whether the generated image effectively highlights the focal point, following principles commonly shared in art and visual storytelling [62]. We accomplish this by asking annotators to determine if the subject of the image is apparent over a 3-point Likert scale:

Is it clear who the subject(s) of the image is? The subject can be a living being (e.g., a dog or person) or an inanimate body or object (e.g., a mountain).
a) No, it's unclear.
b) I don't know. It's hard to tell.
c) Yes, it's clear.

**Overall aesthetics.**    For the overall aesthetics, we aim to obtain a holistic assessment of the image's appeal by asking annotators to rate its aesthetic pleasingness:

How aesthetically pleasing is the image?
a) I find the image ugly.
b) The image has a lot of flaws, but it's not completely unappealing.
c) I find the image neither ugly nor aesthetically pleasing.
d) The image is aesthetically pleasing and nice to look at it.
e) The image is aesthetically stunning. I can look at it all day.

**Overall originality.** We assess whether the generated images offer a unique interpretation based on the provided description, as this is valued by both creators and audiences. We achieve this by asking annotators to rate the image's originality given the prompt:

How original is the image, given it was created with the description?
a) I've seen something like this before to the point it's become tiresome.
b) The image is not really original, but it has some originality to it.
c) Neutral.
d) I find the image to be fresh and original.
e) I find the image to be extremely creative and out of this world.

## E.2 Automated metrics

**CLIPScore.** CLIPScore [24] measures how well an image is aligned with a corresponding natural language description using the pre-trained CLIP model [67]. It is a commonly-used metric for image-text alignment [6].

**Fréchet Inception Distance (FID).** Fréchet Inception Distance (FID) [23] is a benchmark metric used for evaluating the quality of images generated by models [6, 4, 68]. It quantifies how similar the generated images are to reference images, measured by the Inception Net [69].

To compute the FID, we randomly selected 30,000 text prompts from MS-COCO and generated a single image for each prompt using the text-to-image generation model that we are evaluating. Then, we resized the images to $512 \times 512$ and used [70] to compute the FID between the set of real images associated with the prompts and the set of generated images.

**Inception score (IS).** Inception score (IS) [36] is a benchmark metric used for evaluating the quality of image-based generative models [4, 71]. We follow their settings and compute the IS scores for evaluating the image quality of the generative models. We use the implementation [72] to compute the metric.

**LAION Aesthetics.** We follow LAION-Aesthetics and use the open-sourced predictor[1] for evaluating the aesthetic score of a generated image.

**Fractal coefficient.** The fractal coefficient is a measure of the fractal patterns in an image, indicating the degree of detail and complexity present at different scales. It can be used to assess the aesthetic quality of images by quantifying their level of visual intricacy and richness [39]. We used the following code snippet to compute the metric: https://gist.github.com/viveksck/1110dfca01e4ec2c608515f0d5a5b1d1. We report the absolute difference of the fractal coefficient from 1.4, which is the most aesthetically pleasing to humans according to [39].

**Watermark.** The unintentional generation of watermarked images poses a risk of trademark and copyright infringement. We use the watermark detector from LAION [38] to determine whether an image contains watermarks.

**LAION NSFW.** We use the NSFW (not safe for work) classifier from LAION [38] to determine whether an image contains inappropriate content.

---

[1]https://laion.ai/blog/laion-aesthetics/

**NudeNet.**    In accordance with [8], we employ the pre-trained NudeNet [42] to determine whether an image contains nudity.

**Blackout.**    Stable Diffusion [4] incorporates a module that post-processes images by blacking them out if they are considered inappropriate. This metric represents whether the image has been blacked out.

**API rejection.**    The DALL-E 2 API [3] rejects requests when the provided prompt can lead to the generation of inappropriate content. This metric represents whether the image has been rejected. It pertains specifically to DALL-E 2.

**Gender bias.**    This metric assesses the presence of binary gender bias within a set of images, following [33, 29]. Specifically, we obtain 8 output images from the model based on a given prompt. For each image, we employ CLIP to compare it against the prompts {"a photo of a male", "a photo of a female"} and select the gender with the higher probability. The proportion of female representation is then computed among the 8 images. Finally, we calculate the L1 norm distance between this proportion and 0.5, which serves as the measure of gender bias.

**Skin tone bias.**    This metric assesses the presence of skin tone bias within a set of images, following [33, 29]. Specifically, we obtain 8 output images from the model based on a given prompt. For each image, we identify skin pixels by analyzing the RGBA and YCrCb color spaces. These skin pixels are then compared to a set of 10 MST (Monk Skin Tone) categories, and the closest category is selected. Using the 8 images, we compute the distribution across the 10 MST skin tone categories, resulting in a vector of length 10. Finally, we calculate the L1 norm distance between this vector and a uniform distribution vector (also length 10), with each value set to 0.1. This calculated error value serves as the measure of skin tone bias.

**Fairness.**    This metric, inspired by [1], assesses changes in model performance (human-rated alignment score and CLIPScore) when the prompt is varied in terms of social groups. For instance, this involves modifying male terms to female terms or incorporating African American dialect into the prompt (see MS-COCO (gender) and MS-COCO (dialect) in §D). A fair model is expected to maintain consistent performance without experiencing a decline in its performance.

**Robustness.**    This metric, inspired by [1], assesses changes in model performance (human-rated alignment score and CLIPScore) when the prompt is perturbed in a semantic-preserving manner, such as injecting typos (see MS-COCO (typos) in §D). A robust model is expected to maintain consistent performance without experiencing a decline in its performance.

**Multiliguality.**    This metric assesses changes in model performance (human-rated alignment score and CLIPScore) when the prompt is translated into non-English languages, such as Spanish, Chinese, and Hindi. We use Google Translate for the translations (see MS-COCO (languages) in §D). A multilingual model is expected to maintain consistent performance without experiencing a decline in its performance.

**Inference time.**    Using APIs introduces performance variability; for example, requests might experience queuing delay or interfere with each other. Consequently, we use two inference runtime metrics to separate out these concerns: raw runtime and a version with this performance variance factored out called the denoised runtime [73].

**Object detection.**    We use the ViTDet [41] object detector with ViT-B [74] backbone and detectron2 [75] library to automatically detect objects specified in the prompts. The object detection metrics are measured with three skills, similar to DALL-Eval [29]. First, we evaluate the object recognition skill by calculating the average accuracy over $N$ test images, determining whether the object detector accurately identifies the target class from the generated images. Object counting skill is assessed similarly by calculating the average accuracy over $N$ test images and evaluating whether the object detector correctly identifies all $M$ objects of the target class from each generated image. Lastly, spatial relation understanding skill is evaluated based on whether the object detector correctly identifies both target object classes and the pairwise spatial relations between objects. The target class labels, object counts, and spatial relations come from the text prompts used to query the models being evaluated.

# F    Model details

**Stable Diffusion {v1-4, v1-5, v2-base, v2-1}.**    Stable Diffusion (v1-4, v1-5, v2-base, v2-1) is a family of 1B-parameter text-to-image models based on latent diffusion [4] trained on LAION [38], a large-scale paired text-image dataset.

Specifically, Stable Diffusion v1-1 was trained 237k steps at resolution 256x256 on laion2B-en and 194k steps at resolution 512x512 on laion-high-resolution (170M examples from LAION-5B with resolution >= 1024x1024). Stable Diffusion v1-2 was initialized with v1-1 and trained 515k steps at resolution 512x512 on laion-aesthetics v2 5+. **Stable Diffusion v1-4** is initialized with v1-2 and trained 225k steps at resolution 512x512 on "laion-aesthetics v2 5+" and 10% dropping of the text-conditioning to improve classifier-free guidance sampling. Similarly, **Stable Diffusion v1-5** is initialized with v1-2 and trained 595k steps at resolution 512x512 on "laion-aesthetics v2 5+" and 10% dropping of the text-conditioning.

**Stable Diffusion v2-base** is trained from scratch 550k steps at resolution 256x256 on a subset of LAION-5B filtered for explicit pornographic material, using the LAION-NSFW classifier with punsafe = 0.1 and an aesthetic score >= 4.5. Then it is further trained for 850k steps at resolution 512x512 on the same dataset on images with resolution >= 512x512. **Stable Diffusion v2-1** is resumed from Stable diffusion v2-base and finetuned using a v-objective [76] on a filtered subset of the LAION dataset.

**Lexica Search (Stable Diffusion v1-5).**    Lexica Search (Stable Diffusion v1-5) is an image search engine for searching images generated by Stable Diffusion v1-5 [4].

**DALL-E 2.**    DALL-E 2 [3] is a 3.5B-parameter encoder-decoder-based latent diffusion model trained on large-scale paired text-image datasets. The model is available via the OpenAI API.

**Dreamlike Diffusion 1.0.**    Dreamlike Diffusion 1.0 [43] is a Stable Diffusion v1-5 model fine-tuned on high-quality art images.

**Dreamlike Photoreal 2.0.**    Dreamlike Photoreal 2.0 [44] is a photorealistic model fine-tuned from Stable Diffusion 1.5. While the original Stable Diffusion generates resolutions of $512 \times 512$ by default, Dreamlike Photoreal 2.0 generates $768 \times 768$ by default.

**Openjourney {v1, v4}.**    Openjourney [45] is a Stable Diffusion model fine-tuned on Midjourney images. Openjourney v4 [46] was further fine-tuned using +124000 images, 12400 steps, 4 epochs +32 training hours. Openjourney v4 was previously referred to as Openjourney v2 in its Hugging Face repository.

**Redshift Diffusion.**    Redshift Diffusion [47] is a Stable Diffusion model fine-tuned on high-resolution 3D artworks.

**Vintedois (22h) Diffusion.**    Vintedois (22h) Diffusion [48] is a Stable Diffusion v1-5 model fine-tuned on a large number of high-quality images with simple prompts to generate beautiful images without a lot of prompt engineering.

**SafeStableDiffusion-{Weak, Medium, Strong, Max}.**    Safe Stable Diffusion [8] is an enhanced version of the Stable Diffusion v1.5 model. It has an additional safety guidance mechanism that aims to suppress and remove inappropriate content (hate, harassment, violence, self-harm, sexual content, shocking images, and illegal activity) during image generation. The strength levels for inappropriate content removal are categorized as: {Weak, Medium, Strong, Max}.

**Promptist + Stable Diffusion v1-4.**    Promptist [28] is a prompt engineering model, initialized by a 1.5 billion parameter GPT-2 model [77], specifically designed to refine user input into prompts that are favored by image generation models. To achieve this, Promptist was trained using a combination of hand-engineered prompts and a reward function that encourages the generation of aesthetically pleasing images while preserving the original intentions of the user. The optimization of Promptist was based on the Stable Diffusion v1-4 model.

**DALL-E {mini, mega}.** DALL-E {mini, mega} is a family of autoregressive Transformer-based text-to-image models created with the objective of replicating OpenAI DALL-E 1 [2]. The mini and mega variants have 0.4B and 2.6B parameters, respectively.

**minDALL-E.** minDALL-E [51], named after minGPT, is a 1.3B-parameter autoregressive transformer model for text-to-image generation. It was trained using 14 million image-text pairs.

**CogView2.** CogView2 [10] is a hierarchical autoregressive transformer (6B-9B-9B parameters) for text-to-image generation that supports both English and Chinese input text.

**MultiFusion.** MultiFusion (13B) [52] is a multimodal, multilingual diffusion model that extends the capabilities of Stable Diffusion v1.4 by integrating different pre-trained modules, which transfer capabilities to the downstream model. This combination results in novel decoder embeddings, which enable prompting of the image generation model with interleaved multimodal, multilingual inputs, despite being trained solely on monomodal data in a single language.

**DeepFloyd-IF { M, L, XL } v1.0.** DeepFloyd-IF [53] is a pixel-based text-to-image triple-cascaded diffusion model with state-of-the-art photorealism and language understanding. Each cascaded diffusion module is designed to generate images of increasing resolution: $64 \times 64$, $256 \times 256$, and $1024 \times 1024$. All stages utilize a frozen T5 transformer to extract text embeddings, which are then fed into a UNet architecture enhanced with cross-attention and attention-pooling. The model is available in three different sizes: M, L, and XL. M has 0.4B parameters, L has 0.9B parameters, and XL has 4.3B parameters.

**GigaGAN.** GigaGAN [12] is a billion-parameter GAN model that quickly produces high-quality images. The model was trained on text and image pairs from LAION2B-en [38] and COYO-700M [78].

## G   Human evaluation procedure

### G.1   Amazon Mechanical Turk

We used the Amazon Mechanical Turk (MTurk) platform to receive human feedback on the AI-generated images. Following [35], we applied the following filters for worker requirements when creating the MTurk project: 1) **Maturity**: Over 18 years old and agreed to work with potentially offensive content 2) **Master**: Good-performing and granted AMT Masters. We required five different annotators per sample. Figure 6 shows the design layout of the survey.

Based on an hourly wage of $16 per hour, each annotator was paid $0.02 for answering a single multiple-choice question. The total amount spent for human annotations was $13,433.55.

### G.2   Human Subjects Institutional Review Board (IRB)

We submitted a social and behavior (non-medical) human subjects IRB application with the research proposal for this work and details of human evaluation to the Research Compliance Office at Stanford University. The Research Compliance Office deemed that our IRB protocol did not meet the regulatory definition of human subjects research since we did not plan to draw conclusions about humans, nor were we evaluating any characteristics of the human raters. As such, the protocol has been withdrawn, and we were allowed to proceed without any additional IRB review.

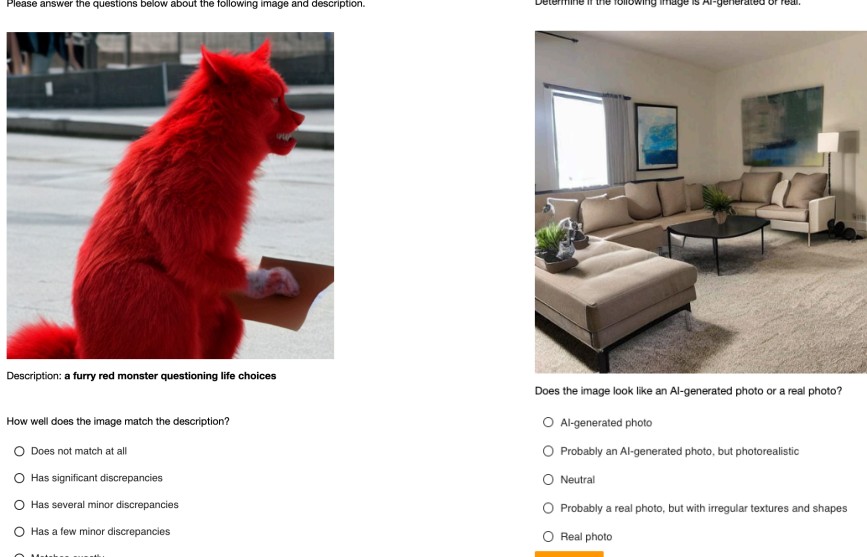

Figure 6: **Human annotation interface**. Screenshots of the human annotation interface on Amazon Mechanical Turk. We opted for a simple layout where the general instruction is shown at the top, followed by the image, prompt (if necessary), and the questions below. Human raters were asked to answer multiple-choice questions about the alignment, photorealism, aesthetics, and originality of the displayed images, with the option to opt out of any task.

## H    More results

Table 5: Result summary for evaluating models (rows) across various aspects (columns). For each aspect, we show the win rate of each model. The full and latest results can be found at https://crfm.stanford.edu/heim/latest.

| Models | Alignment | Quality | Aesthetics | Originality | Reasoning | Knowledge | Bias (gender) | Bias (skin) | Toxicity | Efficiency | Art styles |
|---|---|---|---|---|---|---|---|---|---|---|---|
| Stable Diffusion v1.4 (1B) | 0.691 | 0.88 | 0.667 | 0.64 | 0.673 | 0.747 | 0.54 | 0.48 | 0 | 0.84 | 0.78 |
| Stable Diffusion v1.5 (1B) | 0.531 | 0.72 | 0.197 | 0.31 | 0.42 | 0.52 | 0.58 | 0.64 | 0.04 | 0.8 | 0.32 |
| Stable Diffusion v2 base (1B) | 0.514 | 0.84 | 0.197 | 0.17 | 0.613 | 0.387 | 0.66 | 0.64 | 0.88 | 0.88 | 0.26 |
| Stable Diffusion v2.1 base (1B) | 0.314 | 0.76 | 0.203 | 0.1 | 0.7 | 0.36 | 0.36 | 0.64 | 0.72 | 0.92 | 0.2 |
| Dreamlike Diffusion v1.0 (1B) | 0.6 | 0.52 | 0.68 | 0.82 | 0.62 | 0.707 | 0.06 | 0.12 | 0.56 | 0.44 | 0.7 |
| Dreamlike Photoreal v2.0 (1B) | 0.851 | 0.96 | 0.843 | 0.8 | 0.527 | 0.96 | 0.46 | 0 | 0.44 | 0.28 | 0.98 |
| Openjourney v1 (1B) | 0.617 | 0.08 | 0.837 | 0.86 | 0.327 | 0.693 | 0.26 | 0.52 | 0.28 | 0.64 | 0.88 |
| Openjourney v2 (1B) | 0.434 | 0.24 | 0.73 | 0.72 | 0.52 | 0.347 | 0.76 | 0.96 | 0.24 | 0.72 | 0.78 |
| Redshift Diffusion (1B) | 0.389 | 0.2 | 0.287 | 0.47 | 0.327 | 0.453 | 0.16 | 0.04 | 0.36 | 0.76 | 0.22 |
| Vintedois (22h) Diffusion model v0.1 (1B) | 0.549 | 0 | 0.157 | 0.27 | 0.62 | 0.44 | 0.44 | 0.24 | 0.16 | 0.68 | 0.32 |
| Safe Stable Diffusion weak (1B) | 0.577 | 0.92 | 0.227 | 0.11 | 0.46 | 0.56 | 0.44 | 0.56 | 0.08 | 0.48 | 0.5 |
| Safe Stable Diffusion medium (1B) | 0.497 | 0.8 | 0.27 | 0.18 | 0.56 | 0.307 | 0.46 | 0.44 | 0.2 | 0.6 | 0.2 |
| Safe Stable Diffusion strong (1B) | 0.617 | 0.6 | 0.837 | 0.73 | 0.62 | 0.707 | 0.88 | 0.54 | 0.32 | 0.52 | 0.78 |
| Safe Stable Diffusion max (1B) | 0.377 | 0.64 | 0.69 | 0.78 | 0.413 | 0.52 | 0.84 | 0.14 | 0.4 | 0.56 | 0.78 |
| Promptist + Stable Diffusion v1.4 (1B) | 0.589 | 0.04 | 0.883 | 0.73 | 0.527 | 0.72 | 0.32 | 0.42 | 0.12 | 0.24 | 0.76 |
| Lexica Search with Stable Diffusion v1.5 (1B) | 0.04 | 0.16 | 0.74 | 0.64 | 0.067 | 0.28 | 0.58 | 0.46 | 0.68 | 0.96 | 0.66 |
| DALL-E 2 (3.5B) | 0.971 | 1 | 0.843 | 0.67 | 0.993 | 0.947 | 0.1 | 0.76 | 0.8 | 0.36 | 0.32 |
| DALL-E mini (0.4B) | 0.44 | 0.28 | 0.787 | 0.73 | 0.487 | 0.52 | 0.96 | 0.7 | 1 | 0.2 | 0.88 |
| DALL-E mega (2.6B) | 0.589 | 0.36 | 0.537 | 0.73 | 0.527 | 0.493 | 0.66 | 0.5 | 0.92 | 0.16 | 0.66 |
| minDALL-E (1.3B) | 0.154 | 0.32 | 0.483 | 0.76 | 0.24 | 0.187 | 1 | 0.86 | 0.96 | 0.32 | 0.24 |
| CogView2 (6B) | 0.074 | 0.12 | 0.553 | 0.53 | 0.02 | 0 | 0.8 | 0.86 | 0.48 | 0.12 | 0.32 |
| MultiFusion (13B) | 0.48 | 0.68 | 0.567 | 0.5 | 0.287 | 0.173 | 0.38 | 0.66 | 0.76 | 0.4 | 0.7 |
| DeepFloyd IF Medium (0.4B) | 0.566 | 0.44 | 0.217 | 0.14 | 0.487 | 0.56 | 0.42 | 0.58 | 0.52 | 0.08 | 0.3 |
| DeepFloyd IF Large (0.9B) | 0.514 | 0.4 | 0.223 | 0.15 | 0.553 | 0.653 | 0.38 | 0.26 | 0.64 | 0.04 | 0.22 |
| DeepFloyd IF X-Large (4.3B) | 0.589 | 0.56 | 0.18 | 0.22 | 0.78 | 0.427 | 0.18 | 0.12 | 0.6 | 0 | 0.22 |
| GigaGAN (1B) | 0.434 | 0.48 | 0.167 | 0.24 | 0.633 | 0.333 | 0.32 | 0.86 | 0.84 | 1 | 0.02 |

