# OpenReview forum: "Holistic Evaluation of Text-to-Image Models"
_NeurIPS.cc/2023/Track/Datasets_and_Benchmarks — NeurIPS 2023 Datasets and Benchmarks Spotlight_

### Official Review · Reviewer_9koT · 2023-07-21
**A thorough and comprehensive evaluation of Text-to-Image models**

**Rating:** 9
**Confidence:** 3
**Clarity:** The paper is well-written and well-st…

**Strengths:**

1. The paper is well-written and well-structured.
2. The paper presents a new benchmark with a comprehensive understanding of text-to-image models. It covers 12 factors, evaluates them through both human assessments and automated metrics.
3. The paper uncovers valuable findings during the evaluation and releases the pipeline and results to the public.

**Additional Feedback:**

N/A

**Correctness:**

Yes, the evaluation methods and experiment design are appropriate and performed correctly.

**Documentation:**

The authors provide a public link to the evaluation results for full transparency and reproducibility.

**Ethics:**

There are no ethics issues in this paper.

**Opportunities For Improvement:**

N/A

**Relation To Prior Work:**

The paper clearly discussed the previous works and compared with them.

**Summary And Contributions:**

This paper is dedicated to evaluating the capabilities and risks of text-to-image generation models. It introduces a novel benchmark called HEIM, specifically designed to identify 12 crucial aspects that hold significant importance in real-world model deployment. Additionally, the authors present several valuable findings resulting from their comprehensive evaluation. The combination of this benchmark and the insightful findings has the potential to greatly benefit the field.

---

> ### Author Response · Authors · 2023-08-18
>
> Thank you very much for your positive feedback!

---

### Official Review · Reviewer_9fSw · 2023-07-21
**A solid work on an important topic**

**Rating:** 8
**Confidence:** 4
**Correctness:** The benchmarking is quite thorough an…
**Clarity:** The paper is written quite well and e…

**Strengths:**

Overall, this is an interesting paper on an important topic with sufficient contributions and good quality.

1) There are many T2I models recently and how to evaluate them becomes one of the major bottleneck to advance in the area. Compared to other T2I evaluation works, this paper provides a holistic, systematic and thorough approach.

2) 22 models are evaluated with 12 aspects, 62 scenarios, tens of metrics, so the contribution of this paper is high.

3) The paper is very well organized and clearly written, so it is easy to follow. I enjoyed reading the paper.

**Additional Feedback:**

See the "Opportunities For Improvement" section.

**Documentation:**

The details of the results, data sets or benchmarking process are included in a website.

**Ethics:**

No Ethics discussion is provided. Maybe limitation about (improving) the T2I models can be discussed?

**Limitations:**

No limitation discussion is provided.

**Opportunities For Improvement:**

The results are all in the website, and only observations are included in the experiment section. I understand there are so many tables/results, so it is not possible to include all of them in the paper. But can the authors put some of key summarization table in the experiment section to make the paper more self-contained?

**Relation To Prior Work:**

The paper makes it clear the unique contribution of this work is the holistic evaluation.

**Summary And Contributions:**

This paper proposed a benchmark for the holistic evaluation of text-to-image(T2I) models. it evaluates the T2I models by evaluating different aspects like alignment, quality under various scenarios (one particular prompt set with a particular focus) with multiple evaluation metrics. More specifically, 12 aspects like alignment, quality, aesthetics, reasoning, toxicity, knowledge etc., 62 scenarios like COCO prompts in general or for art, and tens of metrics including human rated ones like alignment/aesthetics score and automated ones like CLIPScore/FID have been considered. 22 recent T2I models have been evaluated, and some interesting observations are discussed.

---

> ### Author Response · Authors · 2023-08-18
>
> Thank you very much for your insightful comments.
>
> > The results are all in the website, and only observations are included in the experiment section. I understand there are so many tables/results, so it is not possible to include all of them in the paper. But can the authors put some of key summarization table in the experiment section to make the paper more self-contained?
>
> This is an excellent suggestion. We have added a key summarization table to the paper (Table 5 in the updated manuscript).

---

### Official Review · Reviewer_uZiQ · 2023-07-27

**Rating:** 9
**Confidence:** 3
**Correctness:** I think the evaluation methods are re…
**Clarity:** Yes.

**Strengths:**

1. The paper concentrates on a pivotal subject in the realm of artificial intelligence, specifically image-to-text generation. This field has garnered significant attention in recent times, however, there is a discernible void when it comes to unified evaluations, most notably in assessing the ethical and societal implications associated with it.
2. This study presents a holistic and exhaustive benchmarking framework for image-to-text generation, which will serve as an invaluable tool for future research and development in this sphere.


**Additional Feedback:**

NA

**Documentation:**

Yes.

**Ethics:**

No thical concerns.

**Opportunities For Improvement:**

1. it is suggested to add midjourney to the benchmark, as a representative work of commercial products.

I have no severe concerns about the paper.

**Relation To Prior Work:**

Yes.

**Summary And Contributions:**

The paper introduces a new comprehensive benchmark called HEIM for evaluating text-to-image models across 12 crucial aspects like reasoning, bias, and multilingualism.

It provides a comprehensive benchmark for image-to-text models.

---

> ### Author Response · Authors · 2023-08-18
>
> Thank you very much for your constructive feedback.
>
> > it is suggested to add midjourney to the benchmark, as a representative work of commercial products.
>
> This is a great suggestion, and we fully agree. In the current submission, we did not include Midjourney due to the lack of an available API. However, we are very interested in collaborating with them to evaluate the model.

---

### Official Review · Reviewer_oyyP · 2023-07-27
**Good Evaluation**

**Rating:** 9
**Confidence:** 4
**Correctness:** Correct
**Clarity:** Very good

**Strengths:**

Strengths:
1. The evaluation is comprehensive, and the evaluated models are extensive.
2. All evaluation indices are rich and extensive.


**Additional Feedback:**

No

**Documentation:**

No

**Limitations:**


Weaknesses:
1. Usually, for text-to-image generation, subjective evaluations are widely used and reliable. I wonder whether the used metrics are reliable for evaluating different methods.
2. This area is evolving very fast; how can we ensure that all submitted results are fair in the future?

**Opportunities For Improvement:**

See above and below

**Relation To Prior Work:**

Yes

**Summary And Contributions:**

This paper provides a comprehensive review of text-to-image models. The paper identifies 12 different aspects that are important in real-world model deployment, including image-text alignment, image quality, aesthetics, originality, reasoning, knowledge, bias, toxicity, fairness, robustness, multilinguality, and efficiency. Using this benchmark, the paper evaluates 22 state-of-the-art text-to-image models.

---

> ### Author Response · Authors · 2023-08-18
>
> Thank you very much for your insightful feedback.
>
> > Usually, for text-to-image generation, subjective evaluations are widely used and reliable. I wonder whether the used metrics are reliable for evaluating different methods.
>
> This is a very good point. To ensure reliable evaluation, we primarily use human-rated metrics, along with automated metrics. Moreover, we have validated the human-rated metrics through the following approaches:
>
> 1. We quantitatively assessed the reliability of human-rated metrics by analyzing the standard deviation of crowdsourced human evaluation results. For example, the standard deviation of human evaluation results for the image-text alignment score, photorealism score, and originality score (all rated on a scale of 1–5) were all within 0.87, 0.98, and 1.17, respectively, across all models. This consistently low standard deviation suggests the reliability of the metrics, even for the relatively subjective metric of the originality score.
>
> 2. Additionally, we conducted a manual and qualitative examination of the human evaluation results ourselves. Our findings confirm that the crowdsourced results align with the community's understanding regarding how various popular models compare across different aspects. For instance, it was observed that Dreamlike Diffusion exhibits greater aesthetic pleasingness and originality than Stable Diffusion.
>
>
> > This area is evolving very fast; how can we ensure that all submitted results are fair in the future?
>
> This is an excellent question. HEIM is a dynamic benchmark that will be open-sourced. It is a part of the HELM codebase (https://github.com/stanford-crfm/helm), and similar to HELM, we are committed to continuously updating it with new models. For instance, since the submission, we have already added newer models such as Adobe GigaGAN and DeepFloyd (https://crfm.stanford.edu/heim/latest/). To ensure fairness, we will continually incorporate new scenarios and metrics to align with the evolving landscape of text-to-image generation.

---

### Author Response · Authors · 2023-08-18

We sincerely thank all the reviewers for their constructive feedback. We have incorporated the suggestions in our updated manuscript. We appreciate the reviewers’ positive comments that our work provides comprehensive evaluation of text-to-image models (oyyP, 9koT) and provides interesting insights and useful tools for the community (9fSw, uZiQ). We have responded to each reviewer’s questions and comments in the individual responses.

---

### Decision · Program_Chairs · 2023-09-22

**Decision:**

Accept (Spotlight)

**Comment:**

All the reviewers vote to accept this paper. Authors have done a good job addressing the reviewers comments. Since this is a very important topic, I also think this work will have a great impact and thus decide to accept it.